# Implementation of a simple thermodynamic sea ice scheme, SICE version 1.0-38h1, within the ALADIN-HIRLAM numerical weather prediction system version 38h1

Yurii Batrak[1], Ekaterina Kourzeneva[2], and Mariken Homleid[1]

[1]Development Centre for Weather Forecasting, Norwegian Meteorological Institute, Oslo, Norway
[2]Finnish Meteorological Institute, Helsinki, Finland
*Correspondence to:* Yurii Batrak (yurii.batrak@met.no)

**Abstract.** Sea ice is an important factor affecting weather regimes, especially in polar regions. A lack of its representation in numerical weather prediction (NWP) systems leads to large errors. For example, in the HARMONIE-AROME model configuration of the ALADIN-HIRLAM NWP system, the mean absolute error in 2 metre temperature reaches 1.5 °C after 15 forecast hours for Svalbard. A possible reason for this is that the sea ice properties are not reproduced correctly (there is no prognostic sea ice temperature in the model). Here, we develop a new SImple sea iCE scheme (SICE) and implement it in the ALADIN-HIRLAM NWP system in order to improve the forecast quality in areas influenced by sea ice. The new parameterization is evaluated using HARMONIE-AROME experiments covering the Svalbard and Gulf of Bothnia areas for a selected period in March – April 2013. It is found that using the SICE scheme improves the forecast, decreasing the value of the 2 metre temperature mean absolute error on average by 0.5 °C in areas that are influenced by sea ice. The new scheme is sensitive to the representation of the form drag. The 10 metre wind speed bias increases, on average, by 0.4 m s$^{-1}$ when the form drag is not taken into account. Also, performance of SICE in March – April 2013 and December 2015 – December 2016 was studied by comparing modelling results with the sea ice surface temperature products from MODIS and VIIRS. The warm bias (of approximately 5 °C) of the new scheme is indicated for the areas of thick ice in the Arctic. Impacts of the SICE scheme on the modelling results and possibilities for future improvement of sea ice representation in the ALADIN-HIRLAM NWP system are discussed.

## 1 Introduction

Sea ice, permanent or seasonal, covers large areas of the ocean, especially in polar regions. Sea ice is a complex system with many important processes occurring. Being an interface between the atmosphere and the underlying medium, the sea ice surface temperature (in contrast to the sea surface temperature) has a noticeable diurnal cycle. Snow accumulation on the ice is accompanied by specific processes of snow-ice formation during the cold season, and by snow melt and the appearance of melt ponds during the warm season. Freezing of saline water results in brine droplets becoming trapped in the ice. This affects not only the ice thermal properties but also the ice structure, due to the slow movement of the trapped droplets towards the ice bottom and the formation of channels. Finally, the ice covered area is not a solid shield but a mixture of floes and

polynyas that drifts being forced by the wind and ocean currents. Large scale ice covered areas strongly affect the properties of the atmospheric surface boundary layer over them. Night time radiative cooling of the ice surface may lead to a very stable boundary layer and limited turbulent exchange between the surface and the atmosphere. Over regions containing a mixture of ice floes and polynyas, the turbulent fluxes are affected by form drag. Thus, it is very important to reproduce these processes over the sea ice correctly in numerical weather prediction models.

Simple parameterization schemes for sea ice are traditionally used in NWP applications. Information about the presence of sea ice cover is taken from observations (the analysis), and the sea ice thickness and sea ice temperature are modelled by a parameterization scheme. For parameterization of sea ice in NWP, two main approaches currently exist: sea ice schemes based on the solution of the heat diffusion equation with several ice layers, but constant ice thickness, e.g. IFS-HRES (Integrated Forecasting System–High RESolution) by ECMWF (European Centre for Medium-range Weather Forecasts) prior to version cy45r1 (ECMWF, 2017a) or HIRLAM (HIgh Resolution Limited Area Model) (Unden et al., 2002); and bulk sea ice models with prognostic ice thickness and an assumed linear or polynomial shape of the temperature profile in the ice, e.g. COSMO (Consortium for Small-scale Modeling) by DWD (Deutscher Wetterdienst) (Mironov et al., 2012; Mironov and Ritter, 2004; Mironow and Ritter, 2003). Snow on ice in these schemes is either not represented (e.g. IFS-HRES), or represented parametri- cally via changing the albedo from ice to snow during the melting period (e.g. COSMO, Mironov and Ritter, 2004). Simple sea ice schemes are used for operational forecasting. However, their performance has mainly been studied in general, with minor validations against observations and without comparisons with more advanced ice models.

More advanced ice models have been developed for ice forecasting applications and research purposes, for example CICE (Community Ice CodE, Hunke et al., 2015), GELATO (Global Experimental Leads and ice for ATmosphere and Ocean, Mélia, 2002) and HIGHTSI (HIGH resolution Thermodynamic Snow and Ice model, Cheng and Launiainen, 1998). They are applied in ocean modelling (e.g. Blockley et al., 2014; Dupont et al., 2015), and in coupled ocean-ice-atmosphere systems for research purposes, climate simulations and seasonal forecasting (Brassington et al., 2015; Lea et al., 2015; MacLachlan et al., 2015; Hewitt et al., 2011; Pellerin et al., 2004). In operational NWP they are applied in the global NWP systems to provide medium- range weather forecasts, e.g., in UK Met Office Unified Model (Walters et al., 2017; Rae et al., 2015) or IFS-ENS (Integrated Forecasting System–ENSemble prediction system, ECMWF, 2017b). However, there are a number of reasons that advanced sea ice models are not widely used for short range operational NWP. Firstly, the advanced ice models are computationally expensive. They parameterize in detail many processes that are important for the evolution of the sea ice itself, but are of secondary importance for the description of ice-atmosphere interactions. Secondly, their robustness and numerical stability during coupling with atmospheric models needs more studies within a framework of short range operational NWP systems. Thirdly, they may require advanced methods of data assimilation for their initialization. In NWP research, in addition to coupled systems, advanced ice models may be used for the performance assessment of simple schemes.

Observations of the sea ice properties that are currently used in NWP are very limited and often only indicate the presence of ice. One example is the sea ice concentration product provided by Ocean and Sea Ice Satellite Application Facility (OSI SAF) (Andersen et al., 2012; Breivik et al., 2001). This product uses observations from passive microwave sensors and is included in the OSTIA (Operational Sea surface Temperature and sea Ice Analysis) product (Stark et al., 2007; Donlon et al., 2012). For

using the sea ice concentration observations in an NWP system, these data need to be projected onto an atmospheric model grid, which is usually done during the analysis step. Subsequently, the sea ice concentration governs the sea surface schemes: for the ice-covered part of a grid box, a simple ice model runs, for the ice-free part, the sea surface temperature is kept constant. Other observations of ice properties (e.g., sea ice microwave emissivity as discussed by Karbou et al. (2014)) are rarely used for assimilation in short range NWP systems, and very few of them are used for validation. In ocean modelling systems, sea ice concentration and sea ice drift data from different sources, including remote sensing from passive microwave and visible channels are used (see, for example, Posey et al., 2015; Sakov et al., 2012). Acquisition of the sea ice depth data from active remote sensing is being developed (Tilling et al., 2016), but the latency of these data is not yet acceptable for operational use in short range forecasting. However, these data could also provide information about sea ice properties for research applications of NWP systems.

In the ALADIN-HIRLAM (Aire Limitée Adaptation dynamique Développement InterNational–HIgh Resolution Limited Area Model) NWP system both the sea surface temperature and the sea ice surface temperature remain constant during the whole forecast. These variables are initialised from an external source (for example, from the global ECMWF model IFS) for each forecast cycle. This causes noticeable errors in near surface air temperature forecasts over ice-covered and in ice-surrounded areas, especially for forecasts longer than 24 hours.

This study presents the development of a simple sea ice parameterization scheme for the ALADIN-HIRLAM NWP system. The scheme solves the heat diffusion equation on a vertical grid within a sea ice slab of constant thickness. This level of simplification was chosen as a first step. In the scheme, provision is made to allow coupling with the snow scheme after Boone and Etchevers (2001). Also, in the case of fractional ice cover, the form drag caused by ice floating over the water surface is taken into account following Lüpkes et al. (2012).

The simple sea ice model (parameterization scheme) is checked for sanity and its performance is assessed through a comparison with the off-line sea ice model HIGHTSI (Cheng and Launiainen, 1998). The overall performance of the HARMONIE-AROME (HIRLAM ALADIN Research On Meso-scale Operational NWP in Euromed–Application of Research to Operations at MEsoscale) configuration of the ALADIN-HIRLAM NWP system with the new sea ice parameterization scheme is evaluated against temperature and wind measurements from coastal meteorological (SYNOP) stations and ice surface temperature products derived from measurements by the MODIS (Moderate Resolution Imaging Spectroradiometer) and VIIRS (Visible Infrared Imaging Radiometer Suite) sensors. The scheme improves the forecast verification scores of the ALADIN-HIRLAM NWP system in coastal areas that are influenced by sea ice but overestimates the sea ice surface temperature in the Arctic, where the prescribed constant value of the ice thickness is too small. The experiments and results described in this paper enable better understanding of the forecast errors and uncertainties and provide an advancement in the description of the interactions between sea ice and the atmosphere in NWP.

The paper is organized as follows. In Section 2 the scheme description and an overview of the physical equations are given. Numerical methods to solve the scheme equation are described in Appendix A. Section 3 evaluates the performance of the new scheme by comparison with the thermodynamic sea ice model HIGHTSI, measurements from SYNOP stations and observations from MODIS and VIIRS. In the final section a short summary of the obtained results is given and the

perspectives for further developments are discussed. Fortran source code of the SICE scheme version 1.0-38h1 is provided in the Supplement.

## 2 Description of the sea ice parameterization scheme

The SImple sea iCE scheme (SICE, pronounced "ess ice") is developed for the parameterization of sea ice in NWP to predict the surface temperature of a thick layer of sea ice. The ice thickness is prescribed. No ice melting or ice formation processes are included and the heat flux from water to ice is neglected. Processes of snow-ice formation, which are discussed e.g. by Saloranta (2000), are not represented. The scheme describes only the processes in the ice slab but it can be coupled to a snow scheme that can provide the value of the heat flux on the lower boundary of the snow layer. The prognostic temperature profile in the ice is obtained from the solution of the heat diffusion equation using the heat balance equation and the temperature of water freezing as upper and lower boundary conditions respectively:

$$
\begin{cases}
C\frac{\partial T}{\partial t} = \frac{\partial}{\partial z}\lambda\frac{\partial T}{\partial z} - \frac{\partial Q}{\partial z} & \\
0 = F + \lambda\left.\frac{\partial T}{\partial z}\right|_{z=0} & \text{if } z = 0, \\
T = T_{frz} & \text{if } z = H,
\end{cases}
\tag{1}
$$

where $t$ is time (s); $z$ is depth (m); $C$ is the volumetric heat capacity of ice (W $\cdot$ s m$^{-3}$K$^{-1}$); $\lambda$ is the ice thermal conductivity (W m$^{-1}$K$^{-1}$); $Q$ is the solar radiation flux penetrating through the ice (W m$^{-2}$); $T$ is the ice temperature (K); $T_{frz}$ is the freezing point of sea water (K); and $H$ is the prescribed ice thickness (m). The term $F$ in the second row of Eq. (1) represents the balance of incoming downward and upward heat fluxes:

$$
F = \delta_{H_{snow}}\left[LW\!\downarrow - \varepsilon\sigma T_s^4 - \rho_a c_p c_{\mathrm{H}} V_{\mathrm{N}}\left(\frac{T_s}{\Pi_s} - \frac{T_{\mathrm{N}}}{\Pi_{\mathrm{N}}}\right) - L\rho_a c_{\mathrm{H}} V_{\mathrm{N}}\left(q_{sat}(T_s) - q_{\mathrm{N}}\right)\right] + (1 - \delta_{H_{snow}})G_{snow}
\tag{2}
$$

where $\delta_{H_{snow}}$ is the Kronecker delta:

$$
\delta_{H_{snow}} =
\begin{cases}
1 & \text{if } H_{snow} = 0, \\
& \text{if } H_{snow} \neq 0;
\end{cases}
\tag{3}
$$

and $T_s$ is the ice surface temperature (K) ($T_s \equiv T|_{z=0}$); $LW\!\downarrow$ is the downward longwave radiation flux (W m$^{-2}$); $\varepsilon$ is the surface emissivity; $\sigma$ is the Stefan-Boltzmann constant (W m$^{-2}$K$^{-4}$); $\rho_a$ is the air density (kg m$^{-3}$); $c_p$ is the air heat capacity with constant pressure (W $\cdot$ s kg$^{-1}$K$^{-1}$); $c_{\mathrm{H}}$ is the drag coefficient for heat; $V_{\mathrm{N}}$ is the wind speed (m s$^{-1}$); $\Pi_{\{s,\mathrm{N}\}}$ is the value of the Exner function on the corresponding level; $T_{\mathrm{N}}$ is the air temperature (K); $L$ is the latent heat of sublimation (W $\cdot$ s kg$^{-1}$); $q_{sat}(T_s)$ is the saturation specific humidity near the ice surface (kg kg$^{-1}$); $q_{\mathrm{N}}$ is the specific humidity of air (kg kg$^{-1}$); $G_{snow}$ is the heat flux from snow to ice (W m$^{-2}$); and $H_{snow}$ is the snow thickness (m). Index N denotes a variable at some level in the atmosphere (the lowest atmospheric model level if the scheme is included in an atmospheric model). The right hand side of Eq. (2) is the sum of the longwave part of the radiative balance $LW\!\downarrow - \varepsilon\sigma T_s^4$ and the turbulent fluxes of sensible

$H = \rho_a c_p c_H V_N \left( T_s/\Pi_s - T_N/\Pi_N \right)$ and latent $LE = L \rho_a c_H V_N \left( q_{sat}(T_s) - q_N \right)$ heat in the case of bare ice, or the heat flux from snow to ice in the case when snow is present.

The term $Q$ in the first row of Eq. (1) describes the heat flux from solar radiation penetrating into the ice pack. This heat flux is calculated by using the Bouguer-Lambert law, with an approximation of radiation absorption in the thin layer of the ice following Grenfell and Maykut (1977):

$$Q(z) = \delta_{H_{snow}}(1-\alpha)SW\downarrow i_0 \cdot e^{-k \cdot z} \tag{4}$$

where $\alpha$ is the ice albedo; $SW\downarrow$ is the downward solar radiation flux (W m$^{-2}$); $i_0$ is the fraction of radiation penetrating through the thin layer of the ice and $k$ is the extinction coefficient for the ice (m$^{-1}$), which is parameterized according to values suggested by Grenfell and Maykut (1977). The value $i_0$ parameterizes the vertical inhomogeneity of the ice transparency and is dependent on depth. It is equal to 1 in the uppermost 0.1 m layer of ice, and equal to 0.18 in the lower layers (Grenfell and Maykut, 1977). Note that in the case of snow on ice, the remaining solar radiation that was not absorbed during penetration through the snow pack is assumed to be completely absorbed by the underlying ice surface.

The main prognostic variable of the SICE scheme is the temperature of the ice. Other parameters are either physical constants or taken from the external forcing. For calculation of the ice thermal conductivity and heat capacity we used the following formulations, which represent their dependency on the ice temperature and salinity (Schwerdtfecer, 1963; Feltham et al., 2006; Sakatume and Seki, 1978):

$$C = C_0 - \frac{T_{mlt}(S) - T_{mlt}(0)}{\theta^2} L \tag{5}$$

$$\lambda = \lambda_{bi} - (\lambda_{bi} - \lambda_b)\frac{T_{mlt}(S) - T_{mlt}(0)}{\theta} \tag{6}$$

where, following Bailey et al. (2010):

$$T_{mlt}(S) = 273.15 - 0.0592S - 9.37 \cdot 10^{-6}S^2 - 5.33 \cdot 10^{-7}S^3$$

$$\lambda_{bi} = \frac{2\lambda_i + \lambda_a - 2V_a(\lambda_i - \lambda_a)}{2\lambda_i + \lambda_a + 2V_a(\lambda_i - \lambda_a)}\lambda_i$$

$$\lambda_i = 1.162\left(1.905 - 8.66 \cdot 10^{-3}\theta + 2.97 \cdot 10^{-5}\theta^2\right)$$

$$\lambda_b = 1.162\left(0.45 + 1.08 \cdot 10^{-2}\theta + 5.04 \cdot 10^{-5}\theta^2\right)$$

$$\lambda_a = 0.03 \qquad V_a = 0.025$$

and $C_0$ is the volumetric heat capacity of the fresh ice (W $\cdot$ s m$^{-3}$K$^{-1}$); $T_{mlt}(S)$ is a function of the melting point of the saline ice depending on the salinity (K); $S$ is the ice salinity, parts per thousand; $\lambda_{\{i,b,bi,a\}}$ is the heat conductivity of fresh ice, brine, bubbly ice and air respectively (W m$^{-1}$); $V_a$ is the fractional volume of air in the sea ice; and $\theta$ is the ice temperature in $^{\circ}$C.

In the case of bare ice (no snow), information about the ice albedo is needed to calculate the surface energy balance from Eq. (2). The ice albedo strongly affects the temperature regime of the ice pack. The effects of some processes taking place on the ice surface, such as the effect of melt ponds, may be parameterized through the ice albedo even without their real physical

description. In the SICE scheme several different parameterisations of ice albedo (Perovich, 1996; Parkinson and Washington, 1979; Roeckner et al., 1992) are available. In these parameterizations, albedo is defined as a constant value or as a function of the ice surface temperature. Numerical methods to solve Eq. (1) and Eq. (2) are presented in Appendix A.

The assumption of bare ice is the simplest possible approximation and may give reasonable results. However, such a simple parameterization describes processes on the ice surface covered by snow in a very approximate way. Snow upon the ice serves as an insulating layer with higher albedo and lower thermal conductivity than the underlying ice. For more physically correct simulations the ice scheme should reproduce the processes related to the evolution of snow on the ice surface. The form of the upper boundary condition presented by Eq. (1), which contains the heat flux $F$ from the snow layer to the ice layer, allows easy coupling with an external snow model to represent snow on ice. In our study, we used the snow module ISBA (Interactions Surface Biosphere Atmosphere) Explicit Snow (ISBA ES, Boone, 2000; Boone and Etchevers, 2001) to represent snow processes. In the current version of SICE, when snow pack exists, it always covers the ice part of a grid cell as a layer of uniform thickness.

ISBA ES is a multi-layer snow scheme with prognostic snow water equivalent, snow heat content and snow density. The number of layers may be defined by the user (the default value is 3). The uppermost snow layer is always less than or equal to $0.05\ m$. The scheme explicitly describes the following processes: snow accumulation due to precipitation, heat redistribution, melting processes and snow pack compaction. It also represents processes related to the melt water within a snow layer. The heat diffusion and surface energy balance equations are solved numerically with implicit schemes. The snow module needs information about the atmospheric forcing and the temperature, heat conductivity and thickness of the topmost layer of ice. It predicts the snow variables and provides the flux from the snow pack to the underlying medium. Thus, the coupling between snow and ice schemes is explicit. The snow surface albedo in ISBA ES is calculated through a simple aging scheme, which contains dry- and wet-snow albedo degradation formulations. In this aging scheme, the snow albedo may decrease during the degradation process from its maximum value of 0.85 to a minimum value of 0.5. When applying this scheme over sea ice, a snow albedo minimum value of 0.75 is used following Perovich (1996); Semmler et al. (2012). The ISBA ES scheme parameterizes snow over land surfaces and contains no parameterizations of specific snow-over-ice processes, such as snow-ice formation or evolution of melt ponds.

An atmospheric model may apply a tiling approach for better representation of the surface processes. This means that a model grid cell may contain a mixture of both sea water and ice. In such cases the ice and open water calculations are performed independently and the output flux to the atmosphere is represented by a weighted average of the fluxes from the water and ice parts. In this case, information about the ice concentration may be utilized to obtain the weighting coefficients. Ice concentration is estimated from satellite observations using the analysis procedure. This procedure contains a consistency check between the sea surface temperature and sea ice concentration fields (Stark et al., 2007; Donlon et al., 2012).

Turbulent exchange between the sea ice and the atmosphere is a complex process that is influenced by the morphological features of the ice pack such as the presence of melt ponds, ice topography, ridges and deformations. In the current version of the SICE scheme these complex features are not represented and the ice part of the grid cell is assumed to be a flat surface with uniform characteristics. When the sea ice concentration is less than 100 % (which means a mixture of open water and

sea ice), one more factor influences the turbulent exchange with the atmosphere. This is the form drag, which is caused by the floes floating on the water with their upper edge higher than the water surface (ice obstacles). This subtle effect might be important for NWP systems that use ice concentration data to define the percentage of sea ice in a grid cell. Indeed, the roughness length of water is lower than that of ice, and simple weighted averaging according to the ice concentration values will lead to a decrease of the roughness length (compared to fully ice-covered area), while in nature it should increase. An accurate sea ice scheme should include a parameterization of the drag caused by ice obstacles (the form drag). Such schemes, discussed for example in Lüpkes et al. (2012), usually introduce an additional term in the weighted average, which depends on the ice fraction. The form drag was introduced into the SICE scheme in the following way:

$$C_{d,mean} = \eta C_{d,ice} + (1 - \eta)C_{d,sea} + C_{d,f} \tag{7}$$

where $C_{d,\{ice,sea,mean\}}$ is the drag coefficient over ice, sea and the mean drag coefficient over the grid cell respectively, under neutral conditions; $\eta$ is the fraction of sea ice in the grid cell; and $C_{d,f}$ is the form drag. The form drag term is calculated by using a parameterization suggested by Lüpkes et al. (2012):

$$C_{d,f} = 7.68 \cdot 10^{-3} \left[ \frac{\ln(0.41/z_{0,w})}{\ln(10/z_{0,w})} \right]^2 (1 - \eta)^\beta \eta \tag{8}$$

where $z_{0,w}$ is the roughness length of the sea water surface and $\beta$ is the tuning constant. Parameters of the SICE scheme are summarized in Table 1.

Technically, the SICE scheme was developed as a part of the externalized land and ocean modelling platform SURFEX (Masson et al., 2013). The externalized surface modelling platform SURFEX is a set of models used for the description of different types of surfaces: sea and inland water bodies, soil/vegetation and urban environments. It assumes a tiling approach, distinguishing different surface types within one grid box of an atmospheric model. Each atmospheric model grid box contains some fraction of 4 different surface types (tiles): nature, urban, inland water and sea. Fractions of these tiles are permanent model parameters obtained from land-use maps. For land-use mapping (physiography) SURFEX incorporates the 1 $km$ resolution database ECOCLIMAPII (Faroux et al., 2013). Over a sea tile, in turn, some fraction of sea ice may exist. This fraction is constant during the forecast run, but it changes at the moment of analysis (model initialization) according to the sea ice concentration estimated from observations. Thus, sea ice may be considered a sub-tile (or patch). The functionality of using main tiles is provided by SURFEX, but the possibility to use information about the fractional sea ice was introduced into SURFEX while implementing the SICE scheme. SICE utilizes the standard heat diffusion equation solver from the SURFEX suite. SICE, which will in future contain a more advanced description of the sea ice, provides technical compatibility with the developing versions of SURFEX.

SURFEX provides diagnostic screen level temperature and specific humidity and 10 metre wind speed from the predicted surface state and the atmospheric values (provided that the forcing is given at some upper level or at the lowest level of the host model) using the interpolation-like procedure of Businger et al. (1971).

SURFEX is incorporated into the ALADIN-HIRLAM NWP system to parameterize the underlying surface processes. The ALADIN-HIRLAM NWP system includes the atmospheric model configuration HARMONIE-AROME (Bengtsson et al.,

2017), which is a version of the non-hydrostatic limited area atmospheric model AROME (Seity et al., 2011). A variety of sub-grid scale physical processes are taken into account by the model parametrization schemes. In the ALADIN-HIRLAM NWP system, boundary conditions and some initial conditions are taken from larger scale models, such as IFS or HIRLAM. The ALADIN-HIRLAM NWP system contains a data assimilation system that uses the three-dimensional variational analysis (3DVAR) method for upper air. Data assimilation of the surface variables uses the optimal interpolation method for snow depth and screen level temperature and relative humidity. In the configuration of the system used in this study, variables in the soil are initialized according to the optimal interpolation method described in Mahfouf et al. (2009).

HARMONIE-AROME performs short-term cycles to produce forecasts. Each cycle contains the data assimilation procedure and the model forecast. The background fields for the data assimilation are fields of prognostic variables at the end of the previous model forecast. In the configuration of HARMONIE-AROME used in this study, the length of the cycle was 3 hours. Starting from 0000 UTC and 1200 UTC analysis times, longer forecasts (up to 48 hours) are performed. Each cycle, the sea water surface temperature and the fraction of sea ice are are kept constant during the forecast being interpolated bilinearly from the host model IFS-HRES, with extrapolation by the nearest neighbour method in specific areas such as fjords. The same is done with the initial value of the sea ice surface temperature. In turn, IFS-HRES uses OSTIA data (Donlon et al., 2012) for the sea surface temperature and ice fraction. For the ice surface temperature, IFS-HRES runs its own simple ice model.

Prior to implementation of the SICE scheme into the ALADIN-HIRLAM NWP system (through SURFEX), sea ice was accounted for in a very crude way in the HARMONIE-AROME configuration. The sea ice surface temperature was initialised by values modelled by IFS-HRES. The ice surface temperature remained constant (equal to its initial value) through the whole forecasting period (similar to the sea surface temperature). This introduced large errors, mainly due to the absence of a diurnal cycle over the ice surface. When the SICE scheme is used, its prognostic variables are updated during each cycle as described in the following. If in the grid cell in question the ice cover exists in the background field, the prognostic variables of SICE are kept unchanged (SICE runs freely). Otherwise, in the situation when the new ice is observed according to OSTIA, the initial (analysed) values of the prognostic SICE variables are obtained via extrapolation from the nearest grid cells of the background field, where the ice exists.

## 3   Performance of the sea ice parameterization scheme

The main objective of the SICE scheme is to reproduce the evolution of ice surface temperature because this variable provides an interface between the atmosphere and the underlying surface. Observations of sea ice surface temperature in the area of interest, which may be used to evaluate the performance of the model/parameterization scheme, are limited. Preliminarily, the modelling results from SICE were compared with the results of the well tested sea ice model HIGHTSI (Cheng and Launiainen, 1998) as an overall technical sanity check of SICE and for better understanding its limitations and weaknesses. For the next step, coupled experiments with SICE were performed with the HARMONIE-AROME model configuration (Bengtsson et al., 2017) of the ALADIN-HIRLAM NWP system for the srping period. Since the end of October 2015, SICE has been run operationally within the ALADIN-HIRLAM NWP system version 38h1.2 by Norwegian Meteorological Institute for the AROME-Arctic

domain (Müller et al., 2017a) (in June 2017 the operational system was updated to the ALADIN-HIRLAM NWP system version 40h1.1). Here we present the results of comparisons of SICE with HIGHTSI, the results of coupled experiments for the spring period and one year period validations of operational runs. Results of spring experiments are verified against the screen level temperature and 10 metre wind speed observations from SYNOP stations and compared with the sea ice surface temperature products from MODIS and VIIRS. The operational runs are compared with MODIS and VIIRS observations only.

## 3.1 Preliminary experiments comparing SICE and HIGHTSI results

HIGHTSI is a one dimensional thermodynamic sea ice model, which was developed for research purposes and climate studies. Although HIGHTSI does not contain ice dynamics (unlike CICE and GELATO), it reproduces temperature profiles in the ice with a sufficient level of accuracy (Cheng et al., 2008) and needs a minimal amount of forcing data. The model describes the evolution of ice mass and energy balance and is based on the heat conduction equation, which is solved with an implicit finite difference numerical scheme (Launiainen and Cheng, 1998). Parameterization of snow in HIGHTSI includes processes of snow accumulation from the forcing precipitation, snow melting and refreezing, and snow-ice formation (which in our experiments was switched off).

### 3.1.1 Design of experiments

SICE and HIGHTSI were run in off-line (stand-alone) mode, because HIGHTSI is not coupled with an atmospheric model. As the atmospheric forcing for SICE and HIGHTSI we used the following variables from HIRLAM (Unden et al., 2002) operational forecasts (with a horizontal spatial resolution of 8 km): lowest model level air temperature, wind speed and specific humidity; surface pressure; global downward shortwave and downward longwave radiation fluxes at the surface; rainfall and snowfall rates. Stand-alone experiments were performed for the 12 selected synoptic stations in the Svalbard coastal area. The period of off-line experiments was from August 2011 to June 2012, with a temporal resolution of the forcing data of one hour. For each model, two experiments were performed: a snow-free experiment, and an experiment considering the evolution of snow. For ice albedo, a simple parameterization based on Roeckner et al. (1992) was used. The ice salinity was set to a uniform value of 3 ppt.

In the SICE scheme, the prescribed ice thickness was given a value of 0.75 m, with 4 layers in the ice slab and 3 layers within the snow. HIGHTSI was configured using the default of 20 layers within the ice slab and 10 layers within the snow pack. The first month of the simulations was considered as a spin-up.

### 3.1.2 Results of comparisons

Stand-alone experiments show that ice surface temperatures modelled by SICE and HIGHTSI may differ by more than 5 °C when difference of the ice thickness in the two models is greater than 0.4 m. Thus, to analyse the difference in reproducing the thermal regime in the ice between the two schemes, we consider only the period when difference of ice thickness in SICE and HIGHTSI is less than 0.4 m. In "no-snow" experiments, this period lasts approximately 3 months (from mid-September to

mid-December) and shows that SICE and HIGHTSI tend to produce similar results. In the experiments with the snow schemes included, the evolution of the snow thickness was quite similar in HIGHTSI and SICE. Due to the presence of snow in these experiments, the ice thickness in HIGHTSI was lower. This led to the period when a difference in ice thickness is less than 0.4 m lasting from mid-September to the end of June. When the ice surface is insulated by snow and only the thin snow layer reacts to the atmospheric forcing, the oscillations of the snow surface temperature are very large. Due to this high variability, the snow surface temperature was sometimes 3–5 °C different between the HIGHTSI and SICE experiments. The mean value of the difference between SICE and HIGHTSI surface temperature for all 12 locations in the "no-snow" experiments was 0.71 °C (SICE gave higher values than HIGHTSI) and standard deviation of differences was 1.04 °C. For "snow" experiments the mean difference and standard deviation of differences are -0.46 °C (SICE gave lower values than HIGHTSI) and 1.99 °C respectively.

The results of the stand-alone experiments show that the SICE scheme adequately reproduces the evolution of the ice surface temperature, however, the result is sensitive to the value of the prescribed ice thickness. Thus, the ice thickness may be important even if the main focus of the simulations is the ice surface temperature. Although the approach with prescribed ice thickness is very simplified, it may reproduce the ice surface temperature oscillations of different time scales and serve as a first approximation for the description of the sea ice cover behaviour.

## 3.2 Experiments with the SICE scheme included in the ALADIN-HIRLAM NWP system: validation against meteorological observations

### 3.2.1 Design of experiments with the ALADIN-HIRLAM NWP system

For coupled experiments, the HARMONIE-AROME model configuration (Bengtsson et al., 2017) of the ALADIN-HIRLAM NWP system described in the end of Section 2 was used. For this study, HARMONIE-AROME experiments were performed over two operational domains (see Fig. 1): (A) the AROME Arctic domain, which includes large ice-covered areas in the Arctic ocean, and (B) the MetCoOp domain, where the ALADIN-HIRLAM NWP system with HARMONIE-AROME is run operationally in a cooperation between the Norwegian Meteorological Institute and the Swedish Meteorological and Hydrological Institute[1] (Müller et al., 2017b), and which covers the Scandinavian peninsula and the Baltic sea. Grids over both domains have a horizontal spatial resolution of 2.5 km. Experiments cover the time period from March to April of 2013. The early spring season was chosen for these experiments because during this part of the year the polar night is already over, but it is still cold enough for the sea ice temperature to have a well pronounced diurnal cycle. Five experiments defined for this part of the study are summarized in Table 2. These are: the reference experiment (REF) without the SICE scheme, SICE experiments without and with the ISBA ES snow module (SICE2D-NS and SICE2D-S respectively), SICE experiment with the form drag parameterization included (SICE2D-AD), and the SICE2D-NS-CLIM experiment which uses the model climatology of the ice thickness provided by TOPAZ4 reanalysis (Sakov et al., 2012; Xie et al., 2017). The following SICE configuration was used in the experiments: 4 layers in the ice, the ice albedo was calculated based on Roeckner et al. (1992). For the SICE2D-AD ex-

---

[1]The Finnish Meteorological Institute joined the MetCoOp collaboration in September 2017 and MetCoOp domain was extended towards the east.

periment the coefficient $\beta$ in Eq. (8) was set to a value of 1. The experiments SICE2D-S, SICE2D-AD and SICE2D-NS-CLIM were only run over the Arctic domain. In SICE2D-S, the default 3 layer configuration of the snow scheme was chosen. The first cycle of the SICE2D-S experiment started from the snow-free state and for the next cycles the initial snow fields were taken from the previous cycle's 3 hour forecast. Snow was accumulated from the precipitation during the whole modelling period.

The ice fraction was taken into account in all SICE experiments. The sea ice fraction was the only sea ice variable that was influenced by observations in the analysis procedure.

### 3.2.2 Results of validation against meteorological observations

The relative impact of the SICE scheme in terms of the root mean square errors (RMSE) of the mean sea level pressure, 2 metre temperature and 10 metre wind speed forecasts starting from 0000 UTC for the time period from 1 March 2013 to 30 April 2013

for all Norwegian national weather stations and SYNOP weather stations within the AROME-Arctic domain is summarized in Fig. 2 and Figs. S1-S3. The number of individual SYNOP measurements is approximately 960 for each station. Experiments with SICE compared to the REF experiment show no considerable differences except at coastal stations surrounded by sea ice.

To evaluate the model performance, we selected stations in the area of Svalbard archipelago and stations situated in the coastal area of the Gulf of Bothnia. According to Fig. 2 and Figs. S1-S3, modelling results from these stations show the largest

relative changes in RMSE when using SICE. To emphasize the effects of using SICE we excluded coastal stations that were always surrounded by open sea in March and April 2013 from the comparison. Some of the selected Svalbard stations are located in fjord areas where the forecast is strongly dependent on the quality of the ice fraction field. Due to the low resolution of the original ice fraction data and a crude extrapolation procedure, for some ice-covered fjords only open water existed in the model runs. Stations located in such fjords were also excluded from the comparison. The final set of SYNOP stations

considered for the comparison consists of 7 stations in Svalbard archipelago and 7 stations in the Gulf of Bothnia. Locations of the selected stations are shown on the Fig. 1.

Figures 3, 4, 5 and 6 show the impact of the new sea ice scheme, including the representation of snow on ice and form drag. These figures show the statistics of the forecast errors obtained by sampling the forecasts starting from 0000 UTC during the period of the experiments for the groups of points in the Svalbard and Bothnian areas. The mean forecast error (bias), the root

mean square error (RMSE) and the standard deviation of errors (ESTD) as a function of the forecast lead time for the mean sea level pressure, 2 metre temperature and 10 metre wind speed were calculated for various experiments. Note that statistics for REF and SICE2D-NS in Fig. 5 and Fig. 6 are different because they cover different time periods: March-April of 2013 for Fig. 5 and only March 2013 for Fig. 6.

The main impact of the SICE scheme is seen in the scores for the 2 metre temperature. Figure 3b shows that in REF, over the

Svalbard stations the 2 metre temperature forecasts have a negative bias increasing in absolute value with the forecast length from 0.5 °C up to 2 °C. This evolution is caused by the influence of the surface temperature over the sea (both the open water and ice cases), which remains constant during the whole forecast period in this experiment. For the Bothnian stations (see Fig. 3b, right panel) in REF, the 2 metre temperature mean error has a diurnal cycle. This is because in the REF experiment for a cycle starting at 0000 UTC the ice surface temperature is initialized from IFS-HRES forecast and represents the cold

night time ice surface. This temperature is in good agreement with reality and, for the night time, the bias in the 2 metre temperature is relatively small. After 12 hours of the forecast, during the day time, sea ice grid cells still hold these very low temperatures and that leads to considerable negative bias in the 2 metre air temperature. The situation is illustrated by Fig. 7, which represents the observed values of air temperature for Kemi I lighthouse (WMO No. 02863, station position: 65°25' N;

24°08' E) and the forecast time series of different length and starting time. It shows that for REF, the air temperature can be more than 5 °C lower in the model forecast than in reality.

The sea ice scheme allows the ice surface temperature to evolve in time and improves the 2 metre temperature forecasts. According to Fig. 3b, over Svalbard stations the 2 metre temperature bias for SICE2D-NS and for SICE2D-NS-CLIM is smaller than for REF, it is now positive and has an almost constant value of 1 °C. For the Bothnian stations (see Fig. 3b) the bias in

SICE2D-NS and SICE2D-NS-CLIM still has a diurnal cycle, but now the night-time errors are much smaller, only 1 °C in absolute value. The RMSE and ESTD of the 2 metre temperature forecasts are also considerably smaller in SICE2D-NS and SICE2D-NS-CLIM compared to REF, especially for forecasts longer than 24 hours. For the Svalbard stations (see Fig. 4b) they are more than 4 °C in REF but only 3 °C in SICE2D-NS and SICE2D-NS-CLIM, and for the Bothnian stations these values (see Fig. 5b) are 3 °C and 2 °C (note that for the Bothnian stations the standard deviation of forecast errors also shows a diurnal

cycle). Thus, forecast errors are smaller in SICE experiments compared to REF and show less variation from station to station. The experiment SICE2D-NS-CLIM, compared to SICE2D-NS, gives slightly better results in terms of the 2 metre temperature forecast biases for Svalbard stations and slightly worse for Bothnian stations (see Fig. 3b). The difference in biases between these two experiments is approximately 0.2 °C and might not be statistically significant. This is because the default sea ice thickness of 0.75 m in the SICE2D-NS is very close to the climatology in the coastal areas.

Although the mean sea level pressure is usually controlled by the large scale rather than local processes, a local positive impact for this field is also visible, especially for the Svalbard stations. Figure 3a shows that the positive bias of up to 0.5 hPa of mean sea level pressure forecasts in REF is removed in SICE2D-NS and SICE2D-NS-CLIM. This occurs due to warmer (in general) temperatures in these experiments. In terms of ESTD of the mean sea level pressure forecasts (and since the bias is small, also RMSE), there is no considerable difference between REF, SICE2D-NS and SICE2D-NS-CLIM for the Svalbard

stations (see Fig. 4a). For the Bothnian stations, there is no considerable difference between REF, SICE2D-NS and SICE2D-NS-CLIM experiments for both the mean error (Fig. 3a), ESTD and RMSE (Fig. 5a) of the mean sea level pressure forecasts.

For the 10 metre wind speed, in REF bias is positive, with values between 0.1 and 0.5 $\mathrm{m\,s^{-1}}$. The experiments SICE2D-NS and SICE2D-NS-CLIM have an approximately 0.5 $\mathrm{m\,s^{-1}}$ higher mean error than REF for all forecast lengths (see Fig. 3c, left panel). The source of the larger wind speed in SICE2D-NS and SICE2D-NS-CLIM is the absence of the form drag over

fractional ice in these experiments. In REF the sea-related part of the grid cell may have only two states: either covered by open water or by ice. As a result, in this experiment all of the Svalbard stations are affected by the surrounding compact ice areas and the simulated wind speed at these points depends on the ice roughness length. In SICE2D-NS and SICE2D-NS-CLIM, the stations are surrounded by a mixture of ice and open water. In this case the average drag coefficient for momentum over a grid cell that contains both open water and sea ice is smaller than in REF, since the roughness of a water surface is much lower

than that of ice. This leads to higher wind speeds. Thus, the large positive bias in SICE2D-NS and SICE2D-NS-CLIM is the

effect of averaging the drag coefficients over open water and ice for the sea-related part of the grid cell. A weighted average is applied according to the ice fraction value and the form drag is not taken into account.

In SICE2D-AD, the form drag is taken into account in the SICE scheme. In this experiment, we add the form drag only when calculating the momentum flux. The effect of the form drag term is shown in Fig. 8, displaying the difference between the drag coefficients calculated in an ordinary way and with the additional term. The impact of the form drag is most noticeable in areas near the ice edge, where the ice fraction field has values of around 60 %. This is in agreement with Elvidge et al. (2016), Tsamados et al. (2014) and Lüpkes et al. (2012). In SICE2D-AD the wind speed bias is smaller than in SICE2D-NS and is just slightly larger than in REF, as shown in Fig. 6. This improvement is seen both for the Svalbard and Bothnian stations, although it is more pronounced for the Svalbard stations due to the differences in the ice concentration fields around Svalbard and in the Baltic sea. However, in the SICE2D-AD experiment the sample size (number of forecasts) might be not large enough to make statistically significant conclusions, especially for the group of Bothnian stations, due to the short experiment length. The error statistics for the other fields are not deteriorated in SICE2D-AD compared to SICE2D-NS (not shown).

In the discussion above, we compared the snow-free experiments, where the physical processes over ice are represented very roughly. A more advanced modelling system should also simulate the snow layer on top of the ice pack. In SICE2D-S, the explicit snow scheme ISBA ES is used to represent the snow over ice. The 2 metre temperature forecast errors are larger for SICE2D-S than for SICE2D-NS and SICE2D-NS-CLIM. The bias in SICE2D-S for the Svalbard stations is almost the same as in REF (see Fig. 3b). For the Bothnian stations, the bias in SICE2D-S is smaller than in REF, but still larger than in SICE2D-NS and SICE2D-NS-CLIM. Also, a shift in the diurnal cycle of bias in SICE2D-S compared to SICE2D-NS and SICE2D-NS-CLIM can be seen. This shift is caused by the difference in the thermal resistances of the snow in the SICE2D-S experiment and ice in SICE2D-NS and SICE2D-NS-CLIM experiments. The ESTD and RMSE of the 2 metre temperature forecasts in SICE2D-S is also larger than in SICE2D-NS and SICE2D-NS-CLIM, but it is still smaller than in REF, both for the Svalbard and Bothnian stations, especially for longer lead times (see Fig. 4b and Fig. 5b). These results are in agreement with the off-line experiments. The cold 2 metre temperatures in the SICE2D-S experiment may be caused by different reasons. When conditions in the atmospheric boundary layer are stable, the cold surface becomes decoupled from the atmosphere, and a positive feedback appears, which induces a further drop of the surface temperature. This situation is very difficult to reproduce in modelling. Moreover, model errors may be either positive or negative. This may depend on errors in boundary layer parameterization, radiation fluxes, snow density or precipitation. This complex problem is well explained e.g. in Slater et al. (2001). In atmospheric modelling it is usually called "the stable boundary layer problem", because it appears during the periods of low shortwave radiation, cooling surface and near-surface inversions. In Atlaskin and Vihma (2012) it is shown how this problem appears in different NWP systems. Also, errors in the amounts of snow accumulated by the model may affect the quality of the screen level temperature forecast. For example, a caveat of the current scheme is the absence of the snow-ice formation representation, which could be important in the case of a thick snow layer covering relatively thin ice. Parameterization of these effects would require description of the ice mass balance, which is not implemented in the current version of SICE. In addition, errors in the snow depth and snow water equivalent over the ice are not corrected by the snow data assimilation procedure, as occurs over land.

Validation against coastal SYNOP observations allows the impact of the sea ice temperature evolution to be understood on the local scale, which is the main concern of regional NWP models. However with observations only from coastal stations, we lack understanding of the ice temperature behaviour over large sea ice covered areas.

## 3.3 Comparisons with observations from MODIS and VIIRS

Although comparing different model experiments with data from SYNOP stations could give us an indirect estimate of the performance of the new ice scheme using the forecast scores, it does not provide much information about the actual quality of the representation of sea ice cover. Data from ice mass balance buoys or manned drifting ice stations are valuable sources of in-situ measurements of sea ice, but these data represent local conditions of the sea ice field and only very few of them are located within the AROME Arctic domain. Therefore, the large scale performance of the sea ice scheme could be better

assessed by using remote sensing data.

In the current study the ice surface temperature products from MODIS (Terra and Aqua satellites) and VIIRS (Suomi NPP satellite) sensors were used to verify the performance of the SICE scheme. The satellite observations of sea ice surface temperature (Hall and Riggs, 2015; Tschudi et al., 2017) were retrieved from the archives of the NASA Land Processes Distributed Active Archive Center (LP DAAC). The resolution of the data is approximately 1 km for MODIS swathes and

750 m for VIIRS swathes.

MODIS and VIIRS ice surface temperature products contain gaps due to cloudiness, which decrease the number of valid data points from a single swathe considerably. To reduce uncertainties caused by mismatch between the cloud masks used to generate MODIS and VIIRS products and cloud cover predicted by HARMONIE-AROME, only cloud-free grid cells (both from the point of view of the model and remote sensing data) were considered for comparison with satellite products. Pixels

of MODIS and VIIRS products that were reported as having quality other than 'best' or 'good' by the quality assessment procedure were excluded from the comparison. The study area for each specific date was selected according to the 'closed ice' map provided by OSI SAF ice edge product (Aaboe et al., 2017) to exclude the marginal ice zone, open sea and coastal regions. This ice edge product uses a threshold value of 0.7 for the ice fraction product of OSI SAF to separate 'closed ice'. In our study, modelling results and ice surface temperature products were compared with each other only within this 'closed ice'

zone.

Usually, for validation of the model against observations, the model data are interpolated to the observational points. However, in our case this is impossible, because the resolution of the remote sensing data is finer than of the model grid. Moreover, the locations of the pixels of the remote sensing products are different for two different swathes. We, therefore, first aggregate the remote sensing data on the atmospheric model grid, and then refer the model errors to the locations of model grid boxes.

Also, we used all available swathes with a time stamp within the one hour window for a given forecast lead time. This method complicates the estimates of statistical significance, because the number of individual observations varies a lot depending on availability of swathes, their spatial location and the cloud-covered area. Therefore, here we provide only general statistics, leaving the details for the future studies.

### 3.3.1    Performance of SICE in HARMONIE-AROME experiments

First, we compared the results of model experiments described in Sec. 3.2 with MODIS and VIIRS data. Figure 9 shows the mean bias, RMSE and ESTD of the ice surface temperature forecasts starting from 0000 UTC for REF, SICE2D-NS, SICE2D-S and SICE2D-NS-CLIM experiments averaged over the whole study area calculated from MODIS and VIIRS products depend-
ing on the forecast lead time. The spatial distribution of the mean error and ESTD of the ice surface temperature after 24 hours of forecast started at 0000 UTC in REF, SICE2D-NS, SICE2D-S and SICE2D-NS-CLIM experiments calculated over the experiment period by using MODIS product is shown on the Figs. 10 and 11. Figures S4 and S5 provide the same maps but using VIIRS ice surface temperature product for verification. It can be seen from Fig. 9a that biases of the ice surface temperature forecasts in general are highest for the SICE2D-NS experiment and lowest for SICE2D-S experiment. Biases of
the ice surface temperature from REF lie between the extremes rendered by SICE2D-NS and SICE2D-S. From Fig. 9a, the experiment SICE2D-NS-CLIM gives smaller biases than SICE2D-NS, but larger than REF. However from Fig. 10 we may see that in the polar Arctic, the maximum area averaged ice surface temperature bias is approximately 6 °C in SICE2D-NS, while in SICE2D-NS-CLIM and REF the biases are smaller, approximately 4 °C. From this map, biases in SICE2D-NS-CLIM experiment are much closer to that in REF. Large biases in the SICE2D-NS experiment, when compared to REF, occur because
in this experiment SICE uses a low value of the prescribed ice thickness, 0.75 m against 1.5 m (ECMWF, 2017a), which is used in the ice model of IFS-HRES. Note that the ice thickness in IFS-HRES is set to reproduce the large scale processes rather than local ones. Also, since the MODIS and VIIRS ice surface temperature products provide information only for the clear-sky conditions, they tend to have a cold bias relative to in-situ measurements (see, e.g.,  Hall et al., 2004).

Standard deviations of forecast errors as a function of forecast lead time in Fig. 9c show that these errors in SICE2D-NS
and SICE2D-NS-CLIM have less variation than in REF and SICE2D-S. This indicates that the ice surface temperature in SICE2D-NS and SICE2D-NS-CLIM follows the observed evolution patterns found in the MODIS and VIIRS products better than in REF and SICE2D-S, while in SICE2D-NS the surface temperature is generally higher. The spatial distribution of the standard deviation of forecast errors is represented on Fig. 11. From this figure, ESTD in the experiments SICE2D-NS and SICE2D-NS-CLIM is in general smaller than in the SICE2D-S and REF experiments by approximately 2 °C. In terms of
ESTD, SICE2D-NS and SICE2D-NS-CLIM show the best scores.

The experiment SICE2D-S shows the smallest forecast bias almost without diurnal variation (Fig. 9a and Fig. 10) for MODIS and VIIRS, but high values of the forecast ESTD (Fig. 9c and Fig. 11). This is in agreement with the point comparisons of Sec. 3.2.

All performed HARMONIE-AROME experiments show smaller forecast bias for the inner part of the ice field and large
errors over the ice edge in the Barents sea (Fig. 10). Such a pattern could indicate inconsistency between the ice concentration field in the model and the real structure of the sea ice field. Another possible source of these errors is the inability of the model to represent characteristics of the sea ice in those areas. This situation can be illustrated by Fig. 10d where the high forecast bias values in the South-Western part of the domain are caused by underestimated ice thickness according to reanalysis climatology

in SICE2D-NS-CLIM. Spatial distributions of the forecast bias and standard deviation of errors are in correspondence with area aggregated statistics.

### 3.3.2 Performance of SICE throughout the year

To check the performance of the SICE scheme throughout the year we compared MODIS and VIIRS ice surface temperature
products with results of operational runs. We considered the time period from 1 December 2015 to 1 December 2016.

    In the operational runs, the snow module in SICE is not active, and the prescribed value of the ice thickness is equal to $0.75$ m, with 4 layers in the ice slab. Data from the operational ALADIN-HIRLAM NWP system archive of Norwegian Meteorological Institute for the AROME-Arctic domain are referenced as AA-OPER throughout the text. Without the SICE scheme, the operational configuration of the ALADIN-HIRLAM NWP system would use the ice surface temperature data from IFS-HRES
keeping it constant throughout the forecast. To imitate the "reference" experiment we used these data from the IFS-HRES operational archive. We refer to this dataset as AA-PRESCRIBED throughout the text. AA-OPER and AA-PRESCRIBED datasets have been validated against MODIS and VIIRS ice surface temperature products.

    Figure 12 shows the RMSE of the ice surface temperature as a function of the forecast lead time for each month. These series are calculated over the AROME-Arctic domain for forecasts initialized at 0000 UTC for AA-PRESCRIBED and AA-OPER
using the MODIS product. The same plots for biases are provided in Fig. S6, and the statistics using the VIIRS product in Figs. S7 and S8. Figures 13 and 14 show the monthly spatial distribution of RMSE of the ice surface temperature after 66 hours of forecast for AA-PRESCRIBED and AA-OPER forecasts initialized at 0000 UTC, calculated using the MODIS product.

    It can be seen from Fig. 12 and Figs. S6–S8 that the quality of representation of the sea ice surface temperature varies considerably throughout the year for both AA-PRESCRIBED and AA-OPER. Averaged over the whole territory monthly
biases of the ice surface temperature forecasts are positive, small during end-spring and summer time, and large during autumn and winter time in both datasets. The corresponding RMSEs are also small during end-spring and summer and large during autumn and winter. Variations in RMSE between two different months can reach approximately 15 °C, with 1 °C in July and 15 °C in November (see Fig. 12). Small forecast errors during the summer time occur due to the warm (in general) state of the sea ice during this season and are constrained by the melting temperature of the ice. Large errors in the ice surface temperature
forecasts during the autumn and winter time are caused by deficiencies in the parameterization of the sea ice cover.

    AA-PRESCRIBED tends to show smaller RMSE than AA-OPER for the short lead times, because IFS-HRES parametrizes the sea ice as a layer with $1.5$ m thickness, which is a better approximation than $0.75$ m used in the SICE operational configuration in the inner Arctic, especially for winter months. However, for forecasts longer than 12 hours AA-PRESCRIBED show the same or worse results than AA-OPER. Using the SICE scheme in AA-OPER constrains the growth of RMSE, while
in AA-PRESCRIBED, where the ice surface temperature remains constant, RMSE grows with the forecast lead time. These results indicate that for short forecasts (shorter than 6 hours) the ALADIN-HIRLAM NWP system better represents the sea ice cover when using the initial state provided by IFS-HRES rather than the SICE scheme; but for the forecasts longer than 12 hours, using the ice surface temperature provided by SICE leads to considerably better results.

The spatial distribution of RMSE of the ice surface temperature forecasts from AA-PRESCRIBED and AA-OPER, which is shown on Fig. 13 and 14, supports the conclusions made from the analysis of the area averaged statistics. In the case of AA-PRESCRIBED, the root mean square error has larger values and is less uniform than for AA-OPER during winter, early spring and autumn months. For the summer months errors are similar.

The prescribed sea ice thickness value of 0.75 m in the operational runs of SICE was chosen to provide the best forecast scores in the coastal areas. From the verification results, we see that with this uniform value of the ice thickness the SICE scheme overestimates the sea ice surface temperature over the large ice covered areas in Arctic. This may deteriorate the large scale dynamic simulations in the operational forecast. However, in regional modelling the large scale dynamics are mainly governed by the boundary conditions from a global model thus less influenced by inaccuracies in the representation of the ice
surface temperature.

## 4    Conclusions

A simple thermodynamic sea ice scheme, SICE, was developed to represent sea ice processes in NWP. In this scheme, the temperature profile in the ice is predicted by solving the heat diffusion equation in the slab of ice with a prescribed thickness. The scheme design allows explicit coupling with a snow scheme, via the fluxes and temperature at the snow-ice interface. Also,
the scheme includes the form drag for momentum flux in the surface layer due to ice obstacles in the case of the fractional ice cover.

The scheme was preliminarily tested by comparing it with the sea ice model HIGHTSI (Cheng and Launiainen, 1998) in off-line mode. In the off-line experiments, when the snow module was switched off in both schemes, the difference in the simulated ice surface temperature between SICE and HIGHTSI was small (the difference standard deviation is equal to 1.04 °C), when the
ice thickness modelled by HIGHTSI was approximately equal to that prescribed in SICE. Due to high variability of the snow surface temperature, the difference in surface temperature between the two schemes was larger when the snow module was included, with a difference standard deviation of 1.99 °C. From this comparison the sensitivity of the results to the prescribed value of the ice thickness was noted.

SICE was evaluated in a coupled framework, within the HARMONIE-AROME configuration of the ALADIN-HIRLAM
NWP system to assess the scheme's performance and to study possible errors. Since the end of October 2015, SICE has been running operationally by the Norwegian Meteorological Institute within the ALADIN-HIRLAM NWP system version 38h1.2 for the AROME-Arctic domain. For the spring period, SICE was carefully evaluated from the coupled experiments; for the rest of the year, the operational archives of the Norwegian Meteorological Institute were used. The ice fraction field for both the SICE experiments and in operational runs was provided by the OSTIA product (Stark et al., 2007) via the lower
boundary conditions from the ECMWF model, IFS-HRES. In the reference experiment, the sea ice surface temperature was taken from the IFS-HRES model and remained constant during the forecasting cycle. Coupled experiments were performed for different configurations of the SICE scheme: the snow-free experiment, the experiment with snow on top of the ice and the experiment with the form drag term included. A separate experiment where the ice thickness in SICE is initialized from

the model climatology was also performed. Data from coastal SYNOP stations in the Svalbard and Gulf of Bothnia regions were used for validation in spring experiments. Comparisons against ice surface temperature products from MODIS and VIIRS allowed a study of performance of SICE on the large scale over an entire year to be done.

Verification of coupled experiments against measurements from coastal SYNOP stations showed that the impact of SICE on the 2 metre temperature scores was positive without the snow model, but with the snow model no clear positive impact was seen. For the mean sea level pressure verification scores, a minor positive impact was seen for all SICE experiments. In the SICE experiments without the form drag compared to the reference experiment (which contains no ice fraction representation), a positive 10 metre wind speed bias was noted. This bias was reduced after accounting for the form drag in SICE. However, our conclusion about the impact of the form drag is still preliminary. Also, the form drag term strongly depends on the ice fraction value, thus ice concentration observations of better quality than low resolution passive-microwave data used in OSTIA are desirable. For example, in Posey et al. (2015) it is shown that using high resolution passive-microwave data from Advanced Microwave Scanning Radiometer 2 (AMSR-2) leads to a substantial decrease in model errors.

Comparisons of the model experiments with the satellite ice surface temperature products over the Arctic domain showed that the ice surface temperature forecast has smaller errors in the reference experiment than in the experiments with SICE (with no snow scheme included) for the first 12 hours of forecast. This happens because the prescribed ice thickness in the sea ice parameterization used by IFS-HRES is tuned to reproduce the large scale fields rather than local effects, and it is larger than in SICE. However general patterns of the ice surface temperature field are well captured by SICE and after 24 hours of the forecast, the predicted ice surface temperature in the experiments with SICE shows smaller root mean square error than in the reference experiment. Considering both the forecast bias and standard deviation of errors, the best results were obtained from the SICE scheme experiments with the ice thickness prescribed from the model climatology provided by TOPAZ4 reanalysis (Xie et al., 2017).

Assessment of the SICE performance throughout the year using data from the operational archive showed that forecast errors with SICE are smaller than without SICE during autumn, winter and early spring. In late spring and summer, errors with and without SICE are similar. This happens due to the situation that the ice surface temperature during this season is close to melting point.

The numerical experiments allow us to conclude that SICE can improve forecasts of the HARMONIE-AROME configuration of the ALADIN-HIRLAM NWP system in ice-surrounded areas, especially for forecasts longer than 24 hours. At the moment we recommend its use without snow parameterization for a trouble-proof result. The prescribed ice thickness is an important parameter, and since no estimates of the ice thickness from observations or other sources are used in the current version of ALADIN-HIRLAM NWP system, it should be tuned.

Of course, in the future the scheme itself should reproduce the spatial and temporal inhomogeneity of the ice thickness. Further development will be focused on the physical processes that control the evolution of sea ice, such as ice freezing and melting. Additionally, possibilities to improve the parameterization of snow on sea ice will be studied. More tests on the parameterization of the form drag are planned. The performance of the scheme will be more carefully evaluated with more experiments, for more regions, more seasons and using more data. For example, summer periods in the Arctic region, when ice

is melting, need more validations using SYNOP observations, more remote-sensing observations and observational campaign results if possible. Melt ponds affect the atmosphere mainly through changing radiation fluxes, but they may also influence the modelling results of a whole NWP system, since they lead to higher uncertainty in the ice concentration observations coming from passive microwave remote sensing. The initialization of the ice parameterization scheme and model error corrections
(especially for the snow module) using observations are also of high importance. The possibilities to use more observations and to develop methods to assimilate them, as well as to improve the methods of using existing observations, should be carefully studied.

*Code availability.* SICE is a part of the ALADIN-HIRLAM NWP system, which is not available to the general public. A copy of the ALADIN-HIRLAM NWP system source code can be obtained, for non-commercial research purposes only, from a member institution of
ALADIN or HIRLAM consortium in applicant's country after signing a standardized License Agreement. An extract from the source code of the ALADIN-HIRLAM NWP system version 38h1 that contains only the source code of the SICE scheme version 1.0-38h1 is available in the Supplement.

## Appendix A: Numerical solution

To solve equations Eq. (1) and Eq. (2) numerically, the ice slab of thickness $H$ is divided into $K$ layers of equal thickness,
except for the topmost layer. For the thickness of the topmost layer, the following formulation is used:

$$z_1 = min \left| z^*, \frac{H - z^*}{K - 1} \right| \tag{A1}$$

$$z^* = \begin{cases} 0.05 & \text{if } H \geq 0.2 \\ 0.25 \cdot H & \text{otherwise} \end{cases} \tag{A2}$$

The ice temperature and thermal properties are assumed to be constant within the current layer. Then, according to the implicit Euler numerical scheme, the first row of Eq. (1) may be rewritten for the layer number $j = 1 \ldots K$ as follows (subscripts denote
the layer number $j$, superscripts $^-$ and $^+$ denote the variables at the beginning and at the end of the time step $\Delta t$, $\Delta z_j$ is the thickness of layer $j$)

$$\frac{C_j \Delta z_j}{\Delta t} \left( T_j^+ - T_j^- \right) = \frac{\bar{\lambda}_{j-1}}{\Delta \bar{z}_{j-1}} \left( T_{j-1}^+ - T_j^+ \right) - \frac{\bar{\lambda}_j}{\Delta \bar{z}_j} \left( T_j^+ - T_{j+1}^+ \right) - Q \big|_{z=z_j}^- + Q \big|_{z=z_j - \Delta z_j}^- \tag{A3}$$

where

$$\Delta \bar{z}_j = \frac{\Delta z_j + \Delta z_{j+1}}{2} \quad \text{and} \quad \bar{\lambda}_j = \frac{\Delta z_j \lambda_j + \Delta z_{j+1} \lambda_{j+1}}{\Delta z_j + \Delta z_{j+1}} \tag{A4}$$

This defines a tridiagonal matrix (see Boone (2000) for a detailed description). The skin temperature of ice could be obtained by integrating the first row of Eq. (1) over the topmost layer assuming that the properties of ice are constant within the selected

layer. Thus, combined with the second equation from the system Eq. (1), Eq. (2) and Eq. (4), the equation for the ice temperature within the skin layer can be written as:

$$C_t \frac{\partial T_s}{\partial t} = \delta_{H_{snow}}(R_n - H - LE) + (1 - \delta_{H_{snow}})G_{snow} + \lambda \left. \frac{\partial T}{\partial z}\right|_{z=z_1} \tag{A5}$$

where $C_t \equiv C|_{z=z_1} \cdot \Delta z_1$ is the surface thermal resistance $(\mathrm{W \cdot s\, m^{-2} K^{-1}})$; $\Delta z_1$ is the thickness of the upper layer of ice (m);
$R_n = (1 - i_0 \cdot e^{-k \cdot z_1})(1 - \alpha)SW{\downarrow} + LW{\downarrow} - \varepsilon\sigma T_s^4$ is the radiative balance. The finite differential representation of Eq. (A5) with the implicit Euler scheme gives the upper row of the matrix Eq. (A3). In the case of no snow it reads:

$$\frac{C_t}{\Delta t}\left(T_s^+ - T_s^-\right) = R_n^\pm - H^\pm - LE^\pm - \frac{\lambda}{\Delta \bar{z}_1}\left(T_s^+ - T_1^+\right), \tag{A6}$$

Note that all the fluxes $R_n^\pm$, $H^\pm$, $LE^\pm$ are calculated using the prognostic variables at the end of the time step. For example, in the case of coupling with an atmospheric model, $H^\pm$ can be written as

$$H^\pm = \rho_a^- c_p^- c_H^- V_N^- \left(T_s^+ - T_N^+\right) \tag{A7}$$

For obtaining the future value of $T_N^+$, a procedure known as "implicit coupling" (Best et al., 2004) is used. According to this procedure, the atmospheric variable $X_N^+$ from the lowest model level at the end of the time step can be found from

$$X_N^+ = A_{X,N}^- \cdot F_{X,S}^\pm + B_{X,N}^- \tag{A8}$$

This procedure uses the coefficients $A_{X,N}^-$ and $B_{X,N}^-$ from the implicit numerical solution of the vertical diffusion scheme
from the atmospheric model, and the surface flux $F_{X,S}^\pm$ of the variable $X$. The coupling coefficients in Eq. (A8) are provided by the host model. Term $R_n$ in Eq. (A6) represents the radiative balance and contains the nonlinear term $\varepsilon\sigma T_s^{+4}$, which defines the thermal radiation flux from the ice surface to the atmosphere at time step $t + \Delta t$. Linearization of this term can be done by use of the Taylor series which results in:

$$\varepsilon\sigma T_s^{+4} \approx 4\varepsilon\sigma T_s^{-3}T_s^+ - 3\varepsilon\sigma T_s^{-4} \tag{A9}$$

Then, Eq. (A8) and Eq. (A9) may be applied to transform Eq. (A6) to the form: $T_s^+ - \mathcal{A}_2 T_1^+ = \mathcal{A}_1$. This form is suitable to be the upper row in the tridiagonal matrix represented by Eq. (A3). For the lower boundary condition, the temperature at the bottom of the ice slab (at the bottom of the layer $K$) is equal to the freezing point of the sea water, according to the last equation of system Eq. (1). In this case, the lower row of the matrix represented by Eq. (A3) can be written as:

$$-\frac{\bar{\lambda}_{K-1}}{\Delta \bar{z}_{K-1}}T_{K-1}^+ + \left[\frac{C_K \Delta z_K}{\Delta t} + \frac{\bar{\lambda}_{K-1}}{\Delta \bar{z}_{K-1}} + \frac{2\bar{\lambda}_K}{\Delta z_K}\right]T_K^+ = \frac{C_K \Delta z_K}{\Delta t}T_K^- + \frac{2\bar{\lambda}_K}{\Delta z_K}T_{frz} - Q|_{z=H}^- + Q|_{z=H-\Delta z_K}^- \tag{A10}$$

The resulting system of linear equations may be solved through the Thomas algorithm (Thomas, 1949).

Actual implementation can be found in the source file `src/surfex/SURFEX/simple_ice.F90` available in the Supplement.

*Competing interests.* The authors declare that they have no conflict of interests.

*Acknowledgements.* The authors would like to thank Bin Cheng, Andrew Singleton, Erin Thomas and Laura Rontu for their constructive comments. We are grateful to Malte Müller for preparing the climatological ice thickness data. Comments and suggestions from the two anonymous referees helped to considerably improve our manuscript.

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

**Table 1.** Physical parameters of the SICE scheme and parameters for the numerical solution. All the parameters except the ice salinity may be selected by the user, the range and default values are given

| Parameter | Value and/or reference |
| --- | --- |
| Number of layers in the ice | $[3, 99]$, the default is 4 |
| Number of layers in the snow | $[3, 99]$, the hard-coded default is 3 |
| Ice thickness | 0.75 m |
| Ice thermal properties | after Schwerdtfecer (1963); Feltham et al. (2006) and Sakatume and Seki (1978) |
| Ice salinity | 3 ppt |
| Ice albedo | after Perovich (1996) or Parkinson and Washington (1979) or Roeckner et al. (1992) |
| Radiative transfer within ice | Bouguer-Lambert law, coefficients after Grenfell and Maykut (1977) |
| Freezing point | the default is -1.8 °C |

**Table 2.** Design of experiments: Exp. name – the experiment name, Domain – the experiment domain, Length – the length of the experiment run, Ice cover – "fractional" or "binary" for the ice fraction taken into account or not, respectively, Ice scheme – which sea ice parameterization scheme is used if any, Ice thickness – how the ice thickness is initialized in case of using SICE, Snow scheme – which snow module is used if any, Form drag – whether the parameterization of the form drag used or not.

| Exp. name | Domain | Length | Ice cover | Ice scheme | Ice thickness | Snow scheme | Form drag |
|---|---|---|---|---|---|---|---|
| REF | Arctic | 03-04.2013 | binary | no | | no | no |
| | MetCoOp | 03.2013 | | | | | |
| SICE2D-NS | Arctic | 03-04.2013 | fractional | SICE | uniform, 0.75 m | no | no |
| | MetCoOp | 03.2013 | | | | | |
| SICE2D-S | Arctic | 03-04.2013 | fractional | SICE | uniform, 0.75 m | ISBA ES | no |
| SICE2D-AD | Arctic | 03.2013 | fractional | SICE | uniform, 0.75 m | no | yes |
| SICE2D-NS-CLIM | Arctic | 03-04.2013 | fractional | SICE | climatology, 0.2–2.2 m | no | no |

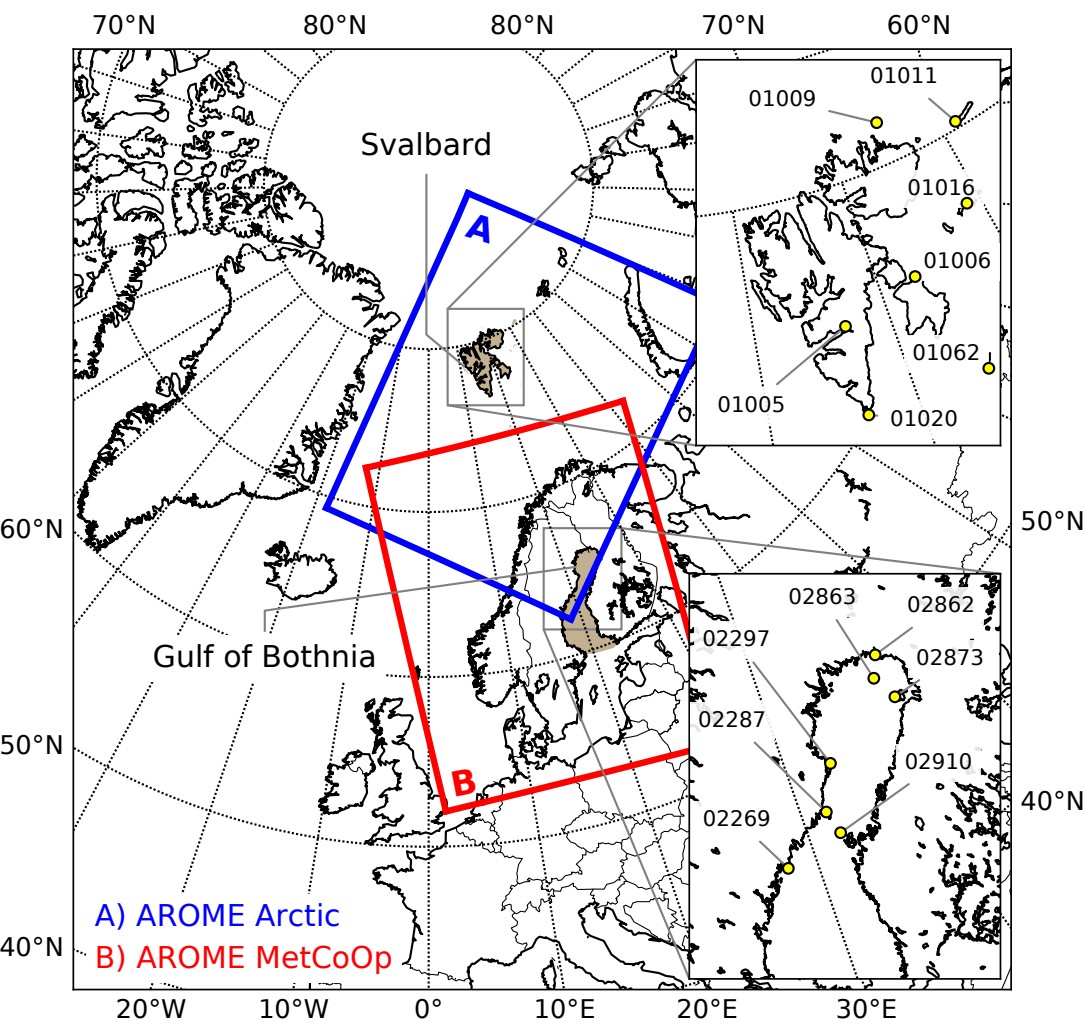

**Figure 1.** Experiment domains. Insets: locations and WMO numbers of the SYNOP stations at Svalbard and around the Gulf of Bothnia used in this study.

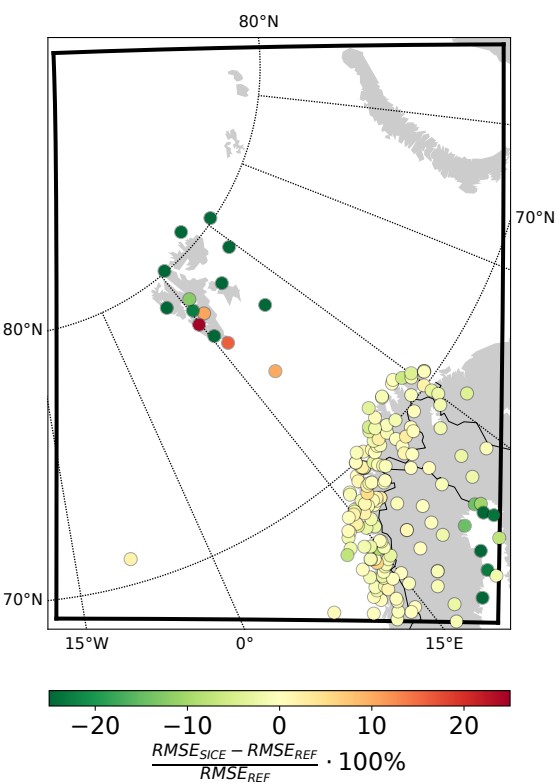

**Figure 2.** Relative change (in percent) of RMSE of the 2 metre temperature forecasts between SICE2D-NS and REF. Negative values mean that the RMSE is smaller in SICE2D-NS than in REF. Forecasts starting at 0000 UTC within the time period from 1 March 2013 to 30 April 2013 were used for comparison.

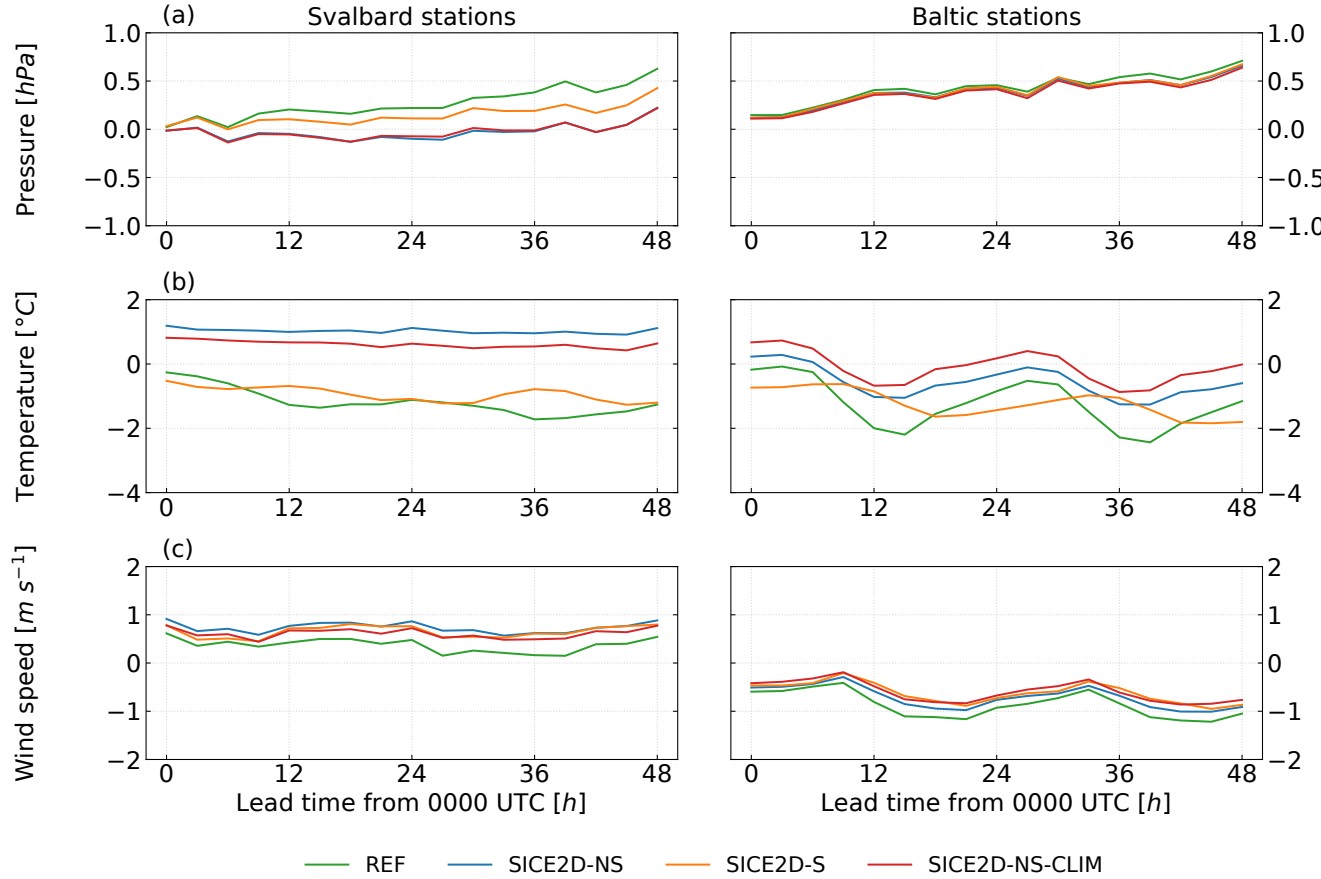

**Figure 3.** Mean error as a function of lead time for forecasts initialized at 0000 UTC for experiments REF, SICE2D-NS, SICE2D-S and SICE2D-NS-CLIM over the AROME Arctic domain for the period from 1 March 2013 to 30 April 2013. Left panel: for 7 Svalbard stations (WMO Nos. 01005, 01006, 01009, 01011, 01016, 01020, 01062). Right panel: for 7 stations in Gulf of Bothnia (WMO Nos. 02269, 02287, 02297, 02862, 02863, 02873, 02910). The mean error is calculated as the forecasted value minus observed value. a) mean sea level pressure; b) 2 metre temperature; c) 10 metre wind speed.

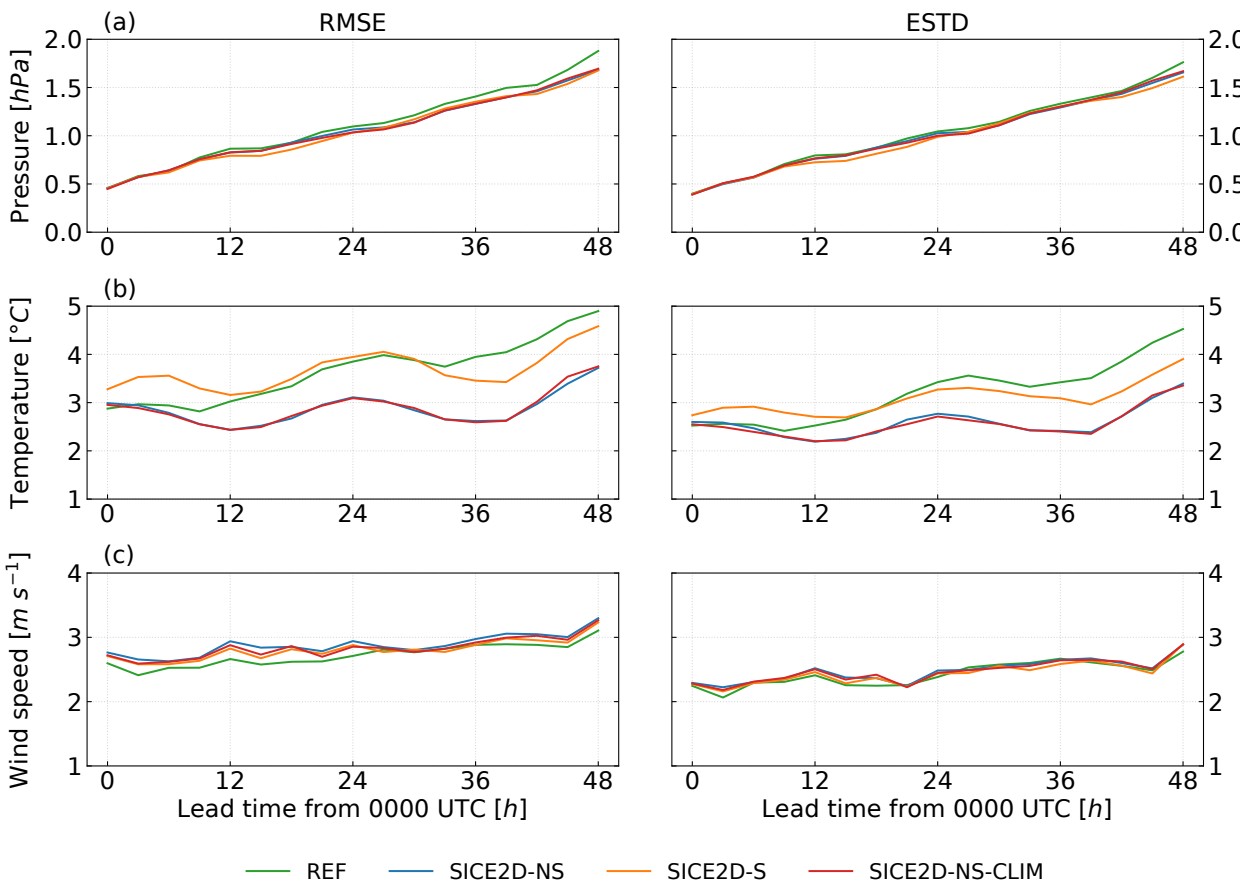

**Figure 4.** Root mean square error (RMSE, left panel) and standard deviation of errors (ESTD, right panel) as a function of lead time for forecasts initialized at 0000 UTC for experiments REF, SICE2D-NS, SICE2D-S and SICE2D-NS-CLIM over the AROME Arctic domain for the period from 1 March 2013 to 30 April 2013. Series are calculated for 7 Svalbard stations. a) mean sea level pressure; b) 2 metre temperature; c) 10 metre wind speed.

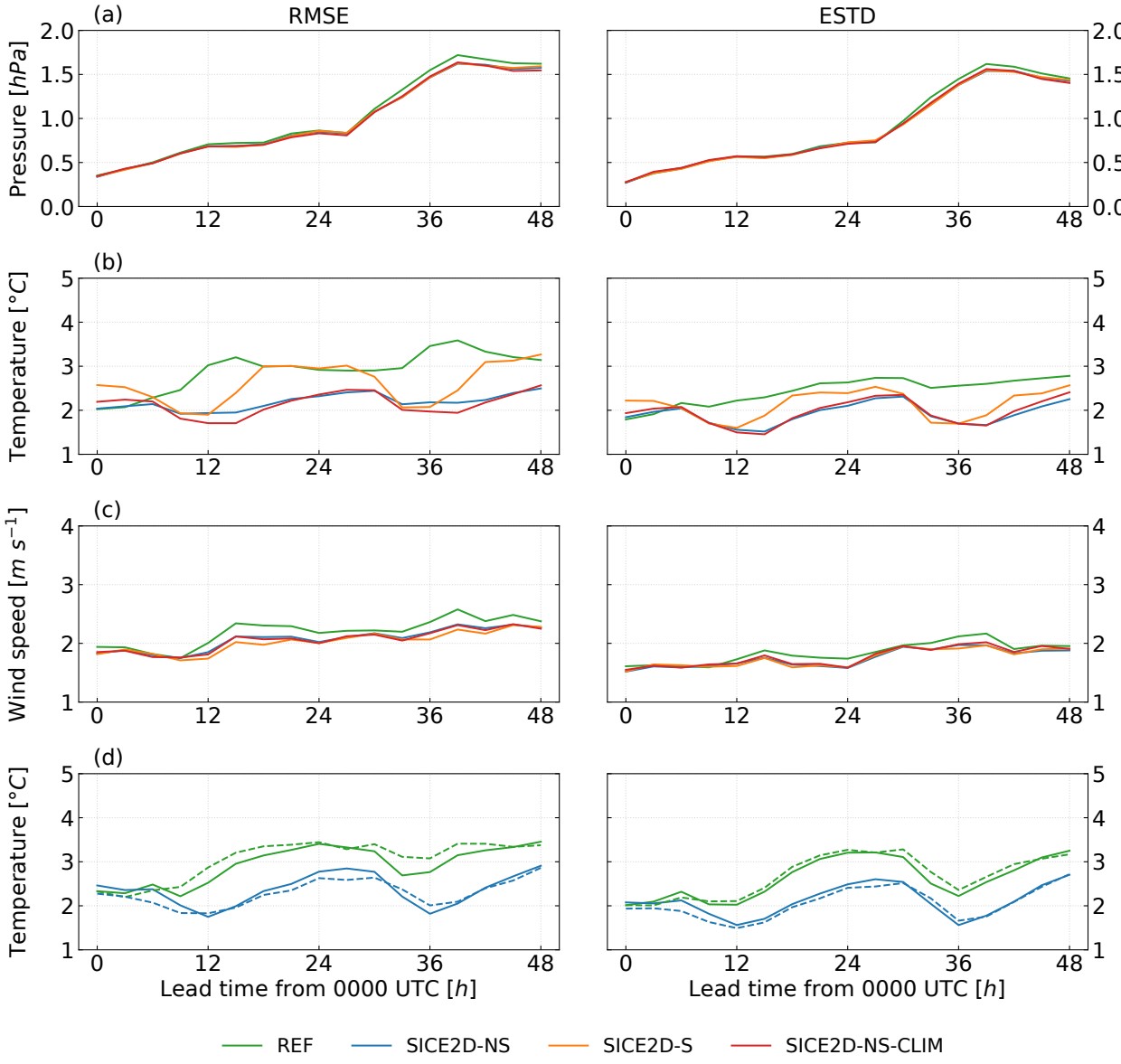

**Figure 5.** Same as Fig. 4, but for 7 stations located in the coastal area of the Gulf of Bothnia. Panel (d) shows the same error statistics as panel (b) but only for March 2013 for experiments REF and SICE2D-NS over AROME Arctic (solid lines) and MetCoOp (dashed lines) domains.

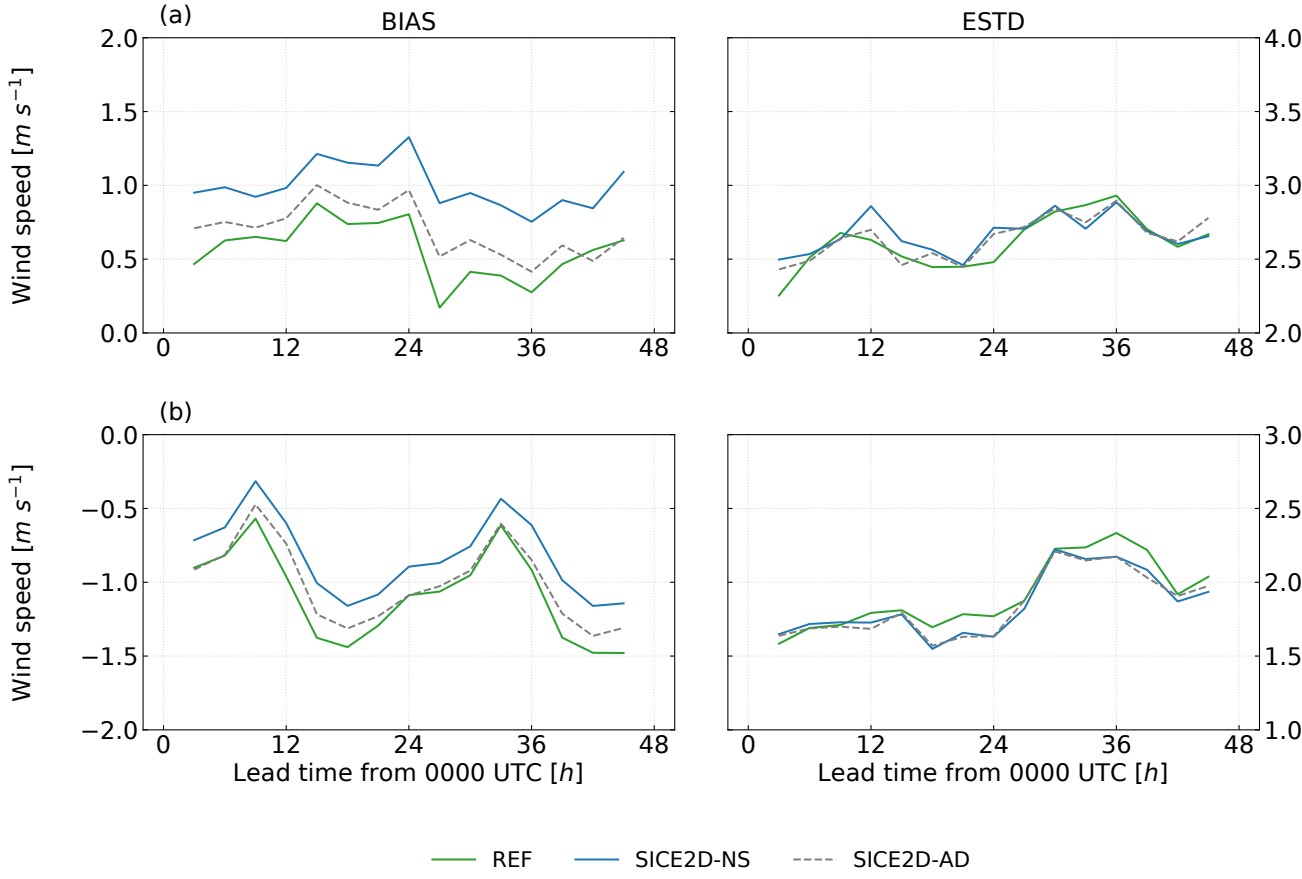

**Figure 6.** Mean error (BIAS, left panel) and standard deviation of errors (ESTD, right panel) of 10 metre wind speed forecasts initialized at 0000 UTC as a function of lead time for the experiments REF, SICE2D-NS and SICE2D-AD over the AROME Arctic domain for the period from 1 March 2013 to 31 March 2013. a) For 7 Svalbard stations; b) for 7 stations located in the coastal area of the Gulf of Bothnia. The mean error is calculated as the forecasted value minus the observed value.

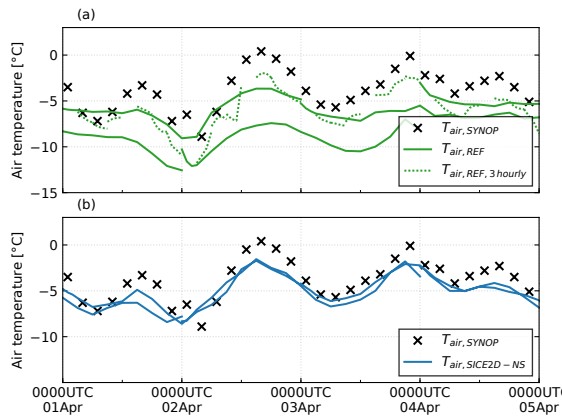

**Figure 7.** Evolution of the 2 metre temperature for the Kemi I lighthouse station (WMO No. 02863, 65°25' N; 24°08' E), observed (black crosses) and simulated by the 48-hour forecasts initialized at 0000 UTC for experiments: a) REF; b) SICE2D-NS. Also in (a), the simulated 2 metre temperature values for the short 3-hour forecasts initialized every 3 hours (except 1200 UTC, which is not shown) that are needed for the initialization of the long forecasts are shown (dotted lines).

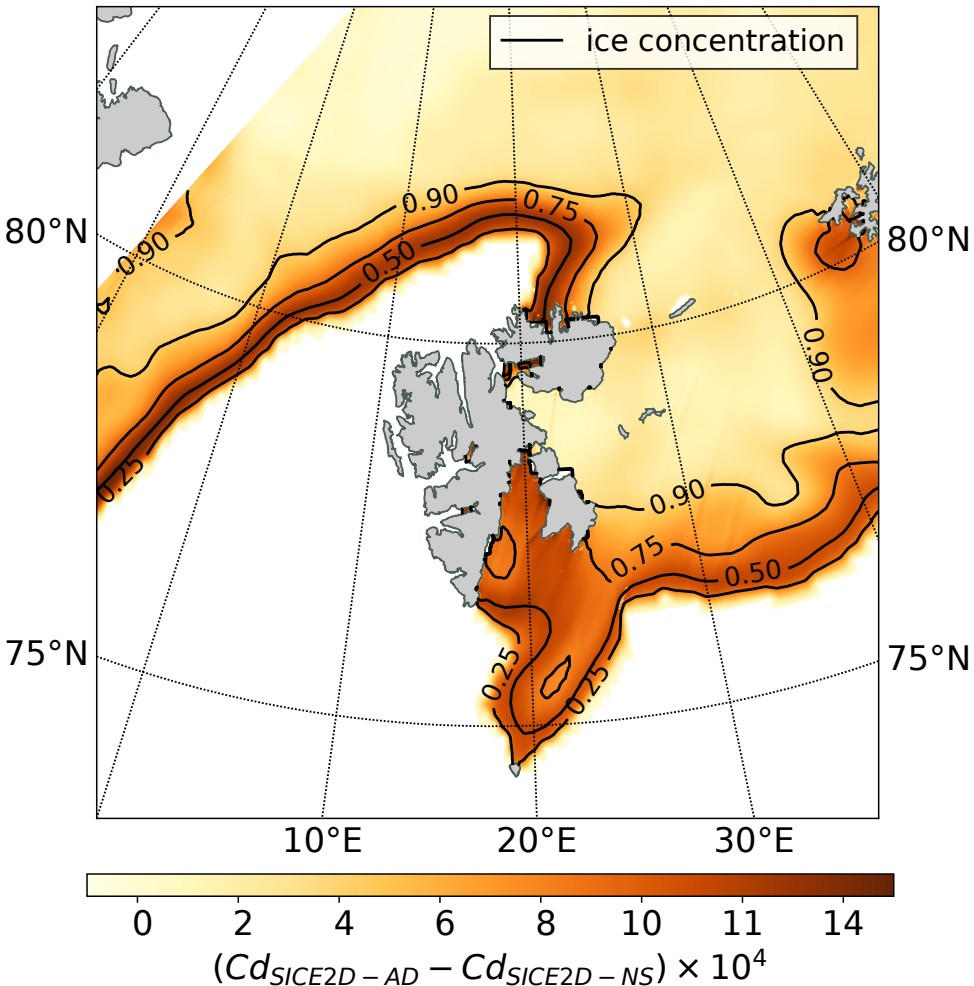

**Figure 8.** Impact of the form drag term on the average drag coefficient. The shading shows the difference between the average drag coefficients over ice covered areas from the SICE2D-NS and SICE2D-AD experiments for 10 March 2013 0000 UTC. Contours show the ice fraction. Open sea and land points are masked.

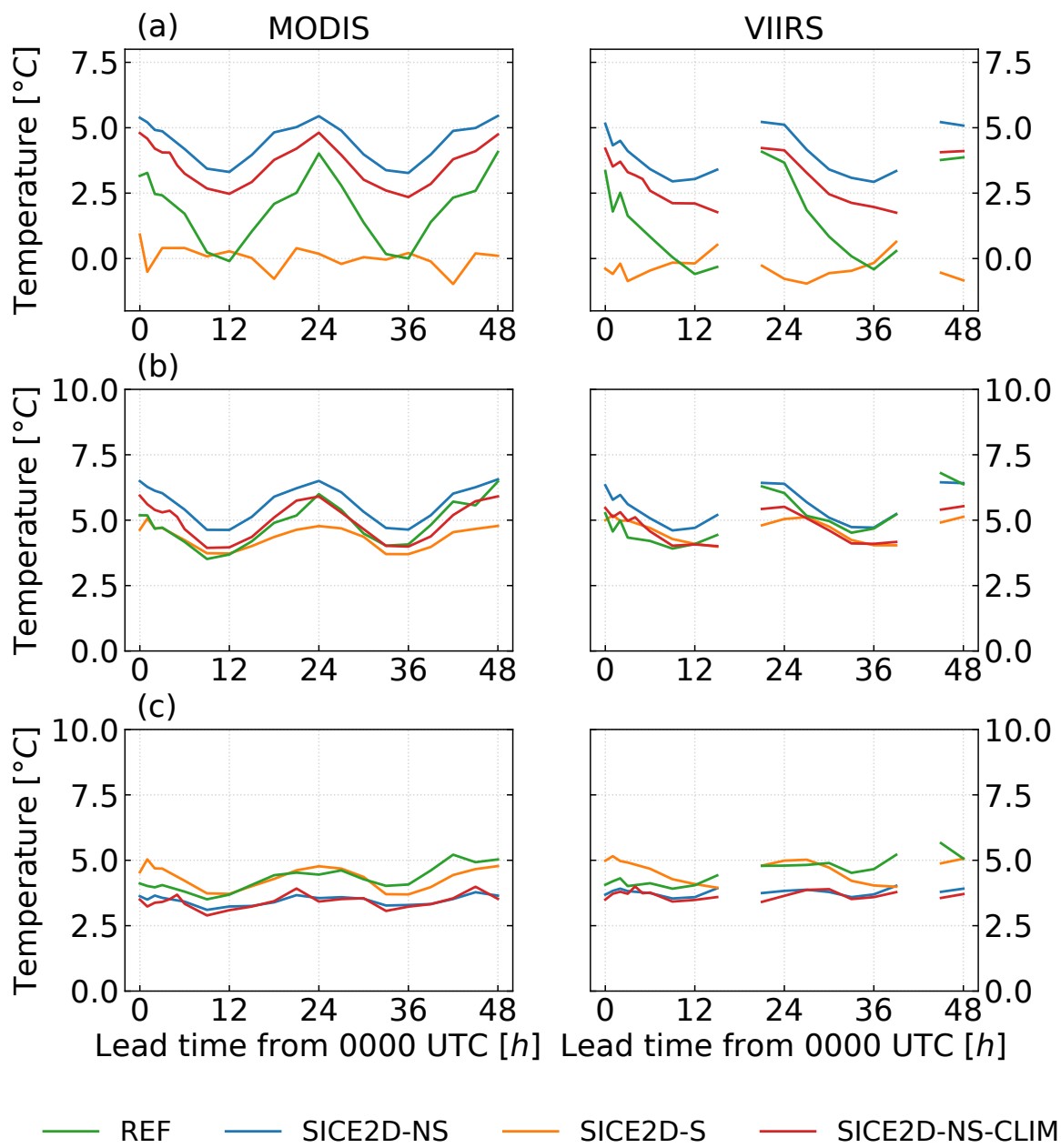

**Figure 9.** Mean bias, root mean square error and standard deviation of errors in the ice surface temperature forecasts initialized at 0000 UTC calculated using MODIS (left panel) and VIIRS (right panel) ice surface temperature products, as a function of lead time for the period from 1 March 2013 to 30 April 2013 for REF, SICE2D-NS, SICE2D-S and SICE2D-NS-CLIM. a) Mean bias; b) root mean square error (RMSE); c) standard deviation of errors (ESTD). VIIRS swathes for the lead times of 18 and 42 hours systematically cover only a minor part of the AROME Arctic domain. They were excluded as not representative.

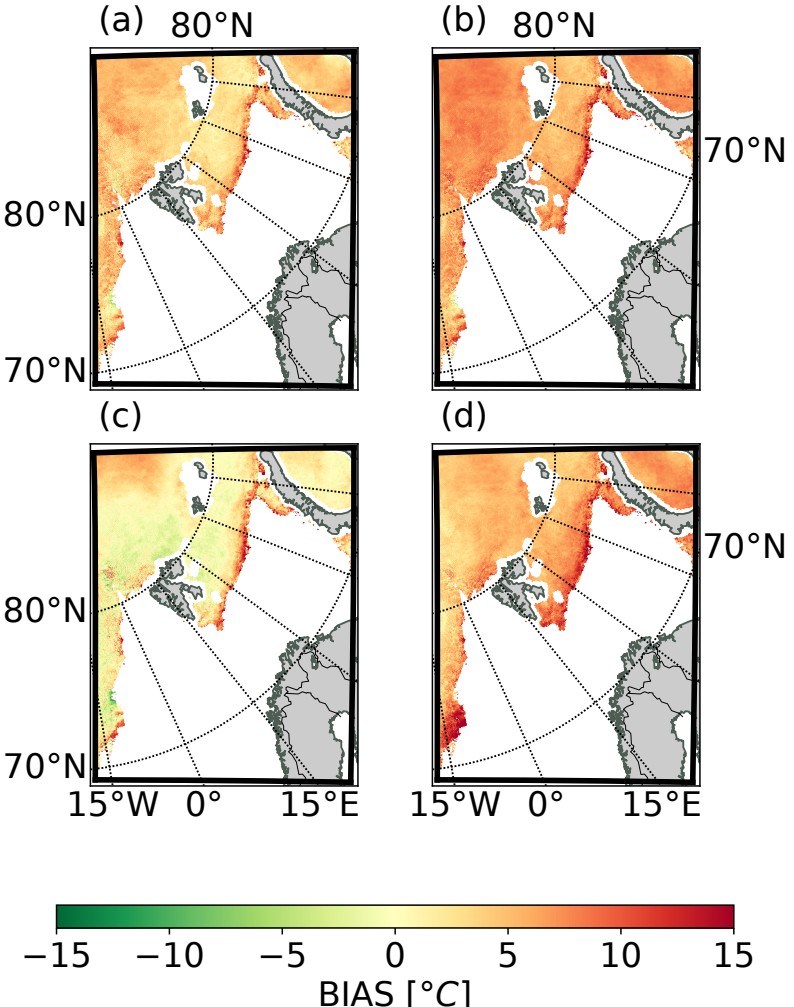

**Figure 10.** Spatial distribution of the mean error of the ice surface temperature after 24 hours of forecast starting at 0000 UTC for REF, SICE2D-NS, SICE2D-S and SICE2D-NS-CLIM compared to the MODIS product. Mean errors are calculated for the time period from 1 March 2013 to 30 April 2013. a) REF; b) SICE2D-NS; c) SICE2D-S; d) SICE2D-NS-CLIM.

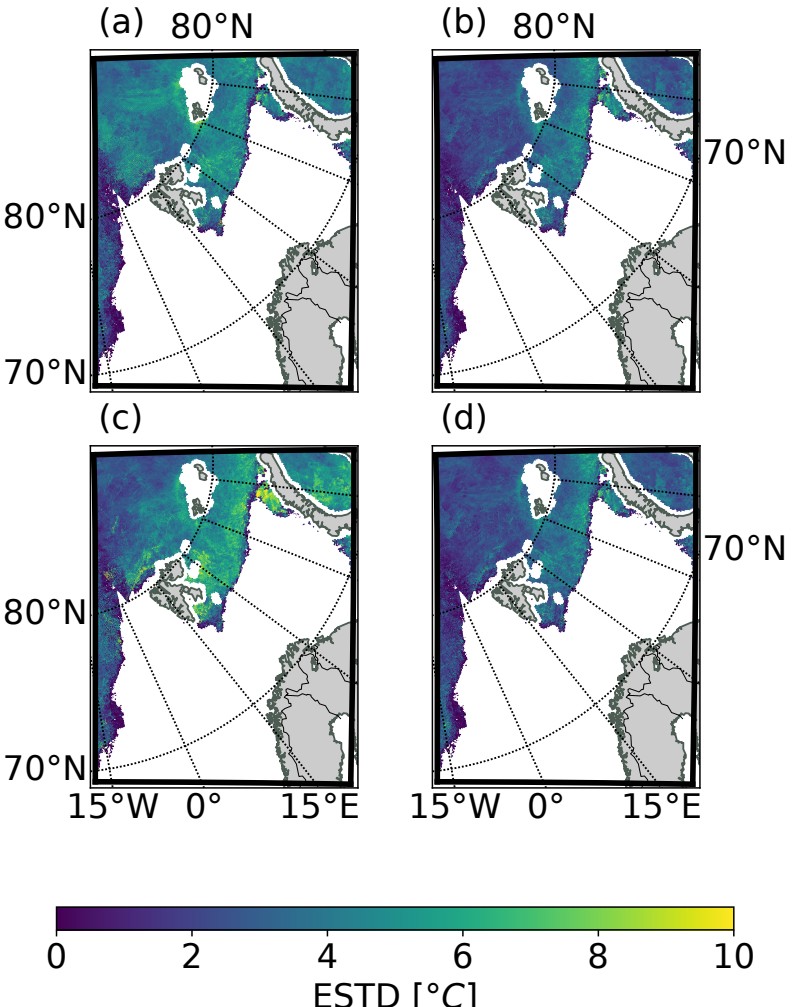

**Figure 11.** Same as Fig. 10 but showing the spatial distribution of the standard deviation of errors (ESTD).

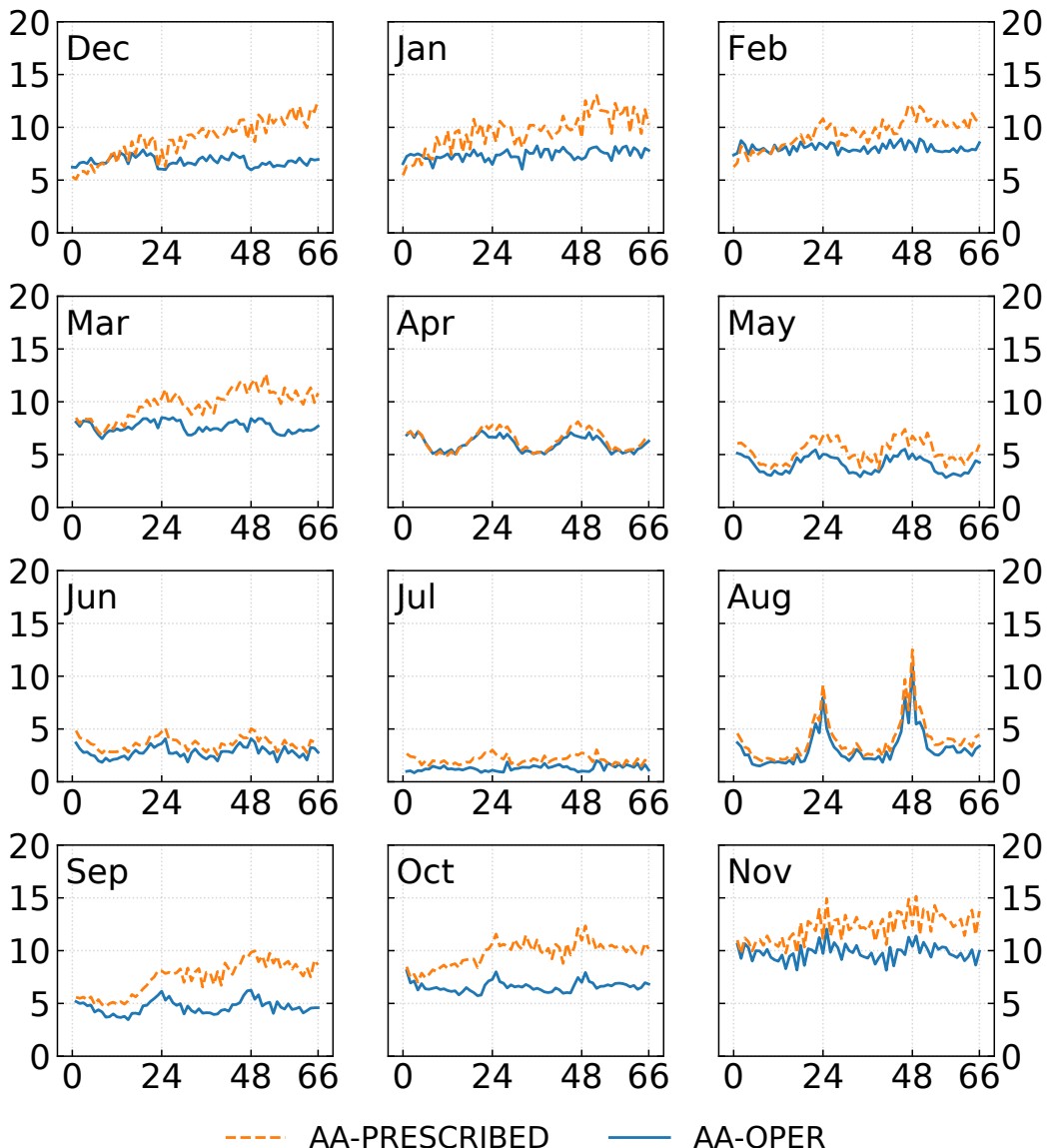

**Figure 12.** Monthly root mean square error (RMSE) of the ice surface temperature as a function of lead time for forecasts initialized at 0000 UTC for AA-PRESCRIBED and AA-OPER (snow-free SICE configuration). Monthly RMSE are calculated using MODIS ice surface temperature product and cover the time period from 1 December 2015 to 1 December 2016. X-axis – forecast lead time from 0000 UTC (h); Y-axis – RMSE of the ice surface temperature (°C).

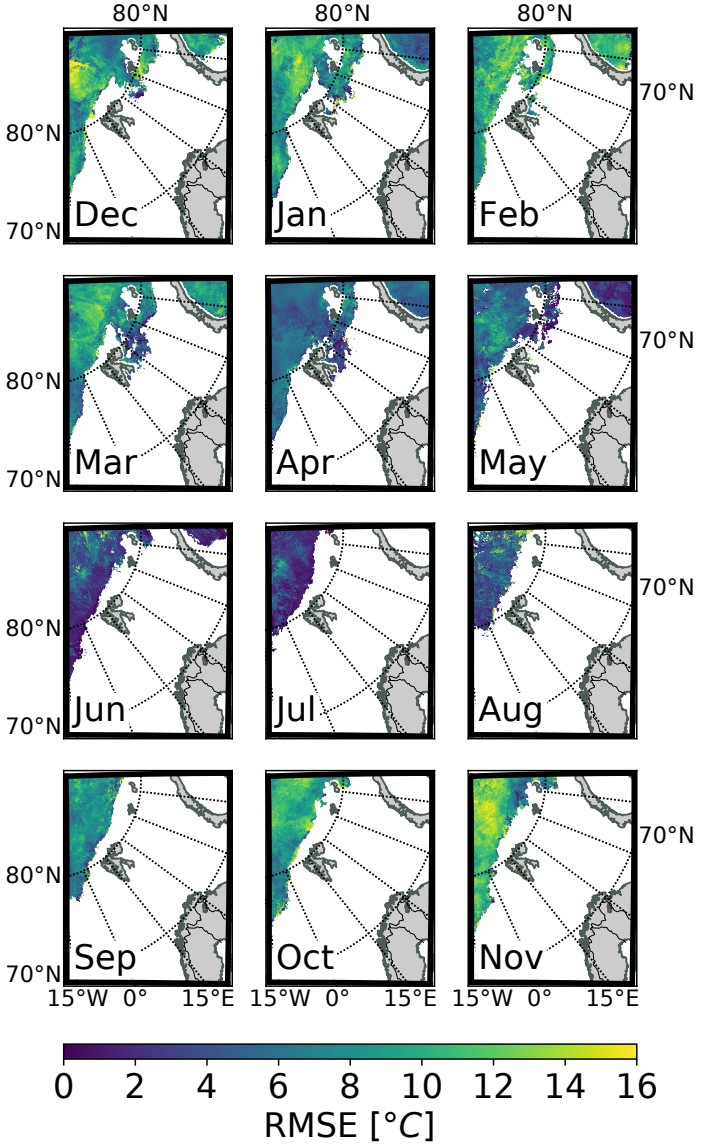

**Figure 13.** Spatial distribution of the monthly root mean square error (RMSE) of the ice surface temperature after 66 hours of AA-PRESCRIBED forecast initialized at 0000 UTC. Monthly RMSE are calculated using MODIS ice surface temperature product and cover the time period from 1 December 2015 to 1 December 2016.

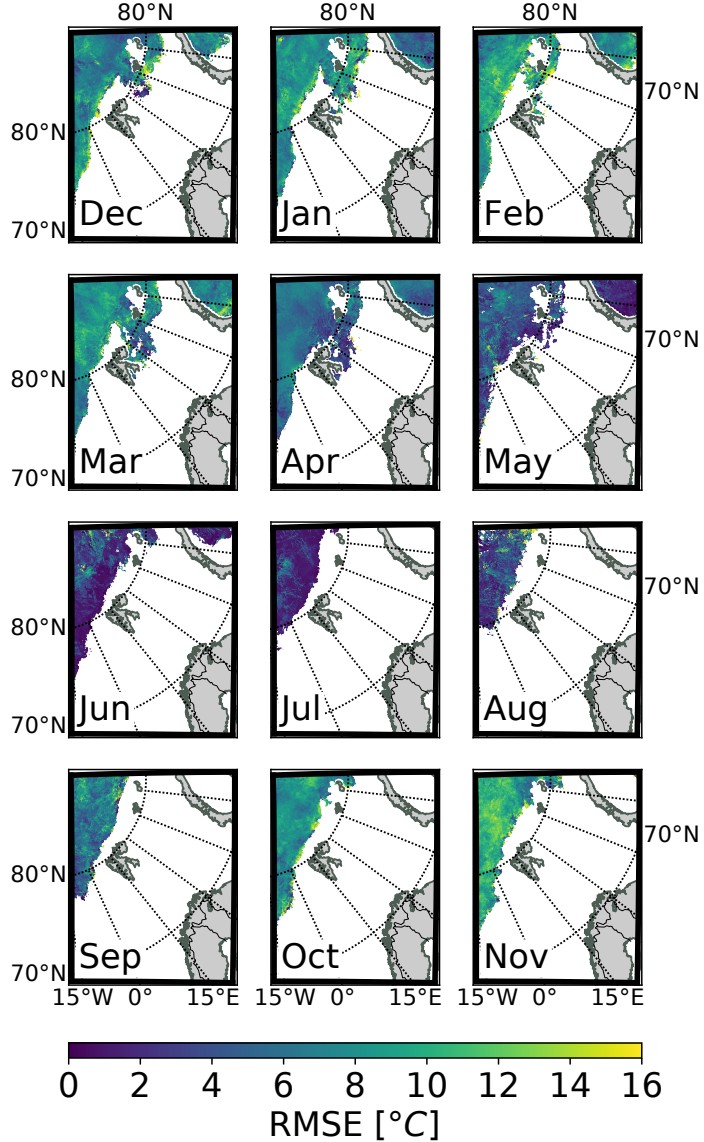

**Figure 14.** Same as Fig. 13, but for AA-OPER (snow-free SICE configuration).