# Peer review of "Implementation of a simple thermodynamic sea ice scheme, SICE version 1.0-38h1, within the ALADIN-HIRLAM numerical weather prediction system version 38h1"

_Geoscientific Model Development, 2018_

## Referee Comment (RC1) · Anonymous Referee #1 · 23 Apr 2018

**1   General comments**

The paper presents results for the implementation of a simple sea ice thermodynamic model in a numerical weather prediction forecasting system. This is the first time such a sea ice model has been implemented in this particular NWP system, and a paper on the implementation and results should thus in principle merit publication.

However, I have serious concerns about the use of a constant ice thickness, about

the limited nature of the results used for validation of the model (covering only a short period in March-April 2013), and about some of the methods used for analysis. Some of the language used is also quite cumbersome and difficult to follow.

I can therefore recommend publication only after major revisions, which I outline below.

**2  Specific comments**

As mentioned above, I thought some of the language used was quite cumbersome and difficult to follow. I would strongly advise the authors to ask a native English speaker to proof-read the paper before resubmitting.

The authors refer in the introduction to the Met Office Unified Model (page 2, lines 19-20), and state that "...to our knowledge there are no publications about the details of coupling between the advanced sea ice model and the atmospheric model in this system". This is incorrect. The model setup for coupled NWP is described by Lea et al. (2015), while the coupling is described in detail by Hewitt et al. (2011). The authors should cite both of these papers.

On page 2, lines 15-18, the authors state that advanced sea ice models are "applied ... in coupled ocean-ice-atmosphere systems for research purposes and seasonal forecasting". I would suggest that they also mention that such sea ice models are used in coupled climate models, such as HadGEM3 (Hewitt et al., 2011).

The description of the model appears to be split between Sections 1, 2 and 3. However, I would prefer to see one single model description section. This could be done in an expanded Section 2. In the current version of the paper, there is some description of the model setup in lines 5-27 of page 3 in the introduction. The authors then describe the sea ice parametrization scheme itself in Section 2. Then, within Section 3, Section 3.2 describes the experimental configuration. I appreciate that some level of

discussion of the model is needed in the introduction, but I think the authors probably go into too much detail here. The introduction should set the scene, describe briefly the scientific background and work done by previous authors, and how the present paper builds on that. Much of the discussion of the model would more properly belong in a model description section. Similarly, Section 3 is primarily a results section. I think the description of the experimental configuration in Section 3.2 would again be better placed in an expanded Section 2.

The model is run with a constant ice thickness of 0.75m. However, the use of a constant thickness is likely to lead to be a source of considerable uncertainty, as ice surface temperature will be extremely sensitive to thickness. Indeed, the authors state in Section 3.1 that "when the ice thicknesses [in SICE and HIGHTSI] are very different, the ice surface temperature values may differ by more than 5°C", and in the conclusions, they say "the sensitivity of the results to the prescribed value of the ice thickness was noticed". Later, they also say that "the simplest way to go forward is to replace the prescribed constant value of the ice thickness by the climatology, to reproduce its seasonal and horizontal large scale variations". This begs the question of why the authors didn't use such a climatology in the current paper. In a resubmitted version of the paper, I would like to see results from a configuration of the model in which the local thickness in each gridbox is prescribed from climatology.

I am confused by Section 3.1. The authors state that they only compared SICE with HIGHTSI where the ice thickness in HIGHTSI was "approximately equal" to the constant thickness used in SICE (0.75m). However, they then state that they consider "small" ice thickness differences to be less than 0.4m, which is more than 50% of 0.75m; I would not say that such thicknesses are "approximately equal" to each other. The authors then state "When the ice thicknesses are very different, the ice surface temperature value may differ by more than 5°C", which suggests that, contrary to their previous statement, they have in fact analysed the results for larger differences in ice thickness between the two models. They also do not state what they mean by "very

different" - do they mean the difference is greater than 0.4m?

In Section 3.2 (page 10, line 16), the authors mention that the model is "started from the snow-free state and allowed to accumulate snow from precipitation during the modelling period". What impact will this have on the forecasts? How long will SICE2D-S take to "spin up" to a realistic representation of the snow cover? It will surely be much longer than the 48 hours of the forecasts in the current paper. For this reason, I am not sure how much weight we can give to the SICE2D-S results. The authors should at least comment on this in the paper, and should preferably present results for SICE2D-S forecasts that have been started from a spun-up snow state.

In Sections 3.3 and 3.4, the authors analyse forecasts only for a short period (March-April 2013). However, the performance of the model is likely to vary during the year, and the results may well be different in different seasons. I note that the authors state in the conclusions that in the future the model will be evaluated for more regions and more seasons, but I am of the opinion that results for other seasons must be included in the present paper for it to be worthy of publication. Given that Arctic sea ice exhibits a clear annual cycle, it is important that the impact of this on the performance of the model is analysed.

In Figures 2, 3 and 6, the authors present results for the mean error in the forecast MSLP, 2-metre temperature, and 10-metre wind speed, where the error is defined as the difference between the modelled and observed quantities, and the mean is taken over several observing sites. However, the standard deviations plotted in Figures 2, 3 and 6 are often much larger than the differences between the means. The authors discuss the differences between the mean errors for different experiments at length in Section 3.3, and consider possible reasons for them, but the fact that the standard deviations are so large compared to the differences suggests that the differences may not be significant. If this is indeed the case, then it suggests that the SICE scheme, and the related snow and form drag schemes, may not have a significant impact on the model-obs errors. It would be interesting to see if this conclusion changes when the

authors use RMS error rather than simple mean error, and when they look at forecasts for different times of year.

Another point relating to Figures 2, 3 and 6 is that the use of a simple mean error will potentially lead to positive and negative errors cancelling each other out. This will hide potentially-relevant results if some stations have a very large positive bias and others have a large negative bias. For this reason, I think the root-mean-square error would be a more useful quantity to assess, and I would like to see a plot of this, either instead of or in addition to the simple mean error that the authors have plotted here.

It would also be interesting to see the contributions of the different observing stations to the mean (or RMS) error. This could be done using maps of a similar form to Figure 3 of Bellouin et al. (2011), where the observations are shown with boxes superimposed on a map showing the fields output by the model.

The authors mention in the text (page 10, line 21) that they used 12 Svalbard stations, and indeed 12 are shown on the map in Figure 1. However, in the captions of Figures 2 and 6, they mention "7 Svalbard stations", and list the 7 stations. I presume this is because of the issues described in Section 3.3 (page 10, lines 23-26) whereby some Svalbard stations were excluded because they were in fjords. But were the other 5 stations used at all in this analysis? If not, then it is incorrect to state at the beginning of Section 3.3 that measurements from 12 Svalbard stations were used (as in fact only 7 were used). The authors should re-word this paragraph (page 10, lines 21-26) to make this clearer.

In Section 3.3 (page 11, lines 14-18), the authors discuss the relative sizes of the standard deviations of the errors in REF and SICE2D-NS, without any mention of the implications or relevance of these results. Presumably a smaller standard deviation implies that there is a smaller range of errors between stations. Is this relevant, and if so why? Is there any indication what might be causing it? Does the implementation of the sea ice scheme affect the 2-metre temperature at some stations more than others?

Is there an obvious reason for this?

At the end of Section 3.3 (page 12, lines 31-34), the authors state "...with observations from coastal stations only, we lack understanding of the ice temperature behaviour for larger scales". This is a very good point to make, and I would recommend that when the authors resubmit the paper they include results for a wider range of stations within the forecast domain, including non-coastal (i.e. inland) stations. Does the implementation of the sea ice scheme affect the results only at stations that are physically close to the sea ice, or are there larger-scale effects?

Figure 7 shows surface temperature derived from MODIS data, and forecast by the model. However, it is quite difficult to get an idea of the differences between the temperature fields in the plots. It would be much more useful if the authors could present maps showing the difference between these (i.e., model minus MODIS). This would help the reader to understand better the results discussed in Section 3.4.

The authors mention in Section 3.4 (page 13, lines 10-12) that most MODIS swaths were in the daytime. However, the model results shown in Figure 7 are whole-day averages. How will this affect the comparison of the two? I imagine there may be a warm bias in the MODIS observations as a result of the fact that they are generally restricted to daytime. The authors should comment on this, and its implications for the results, in the paper.

**3  Technical corrections**

- Page 2, lines 2-3: "Over areas with a mixture of floes and polynyas, the form drag appears, which affects the turbulent fluxes". I would suggest re-wording this, so that it reads: "Over areas with a mixture of floes and polynyas, the turbulent fluxes are affected by form drag".

- Page 2, lines 19 and 34: I don't like the use of "To our knowledge...", as it seems unscientific to me. I have already mentioned above that the statement made on lines 19-20 is in fact incorrect. I would also suggest an alternative wording for the sentence on lines 34-35. Indeed, if one doesn't know whether a particular statement is true or not, it is often best not to include it at all, rather than preceding it with "To our knowledge...".

- Page 4, line 6: "...it is designed so that it can be naturally coupled with a snow scheme...": I don't know what the authors mean by "naturally coupled". I think that "...so that it can be coupled to a snow scheme..." would suffice.

- Page 7, line 25: "It is important to mention that...": This is unnecessary. If it's important to mention it, then mention it – there is no need to say that it's important to do so.

- Page 9, line 25: "The background for the data assimilation are fields of prognostic variables...": I think there is a word missing here, and that this should read " The background fields for the data assimilation...".

- Page 10, line 27: Figure 6 is mentioned before Figures 4 and 5. The figures should be re-ordered to avoid this.

- Page 11, lines 6-7: "...the underestimation of night-time 2 metre temperatures over land is a characteristic feature of the model known from operational verification (not shown)". If this is known from operational verification, is there a reference that the authors can cite?

- Page 11, lines 14-15: "The error standard deviation for the 2 metre temperature forecasts...". This should read "The standard deviation of the errors in the 2 metre temperature forecasts...".

- Page 11, line 22: "mean sea level pressure error standard deviation" sounds clumsy. I would suggest "standard deviation of the error in mean sea level pressure". The authors could also abbreviate "mean sea level pressure" to "MSLP", if they define the abbreviation the first time they use it.

- Page 11, line 31: " over the part of the grid cell related to the sea with ice". I'm not sure what this means. Does it mean "over the ice-covered part of the grid cell", or something else? I would suggest re-wording this to make it clearer.

- Page 12, lines 4 and 18: I think the authors mean "in agreement with" rather than "in accordance with".

- Page 12, line 20: "...makes the surface temperature drop more and more". This language ("more and more") is not very scientific. Please consider re-wording.

- Page 14, line 18: "...the sensitivity of the results to the prescribed value of the ice thickness was noticed". I think the authors mean "noted" rather than "noticed".

- In the caption of Figure 3, the authors mention 7 stations in the Gulf of Bothnia, and 7 are shown in the map in Figure 1, but in the text (page 10, line 21) they state that they used 6 stations in the Gulf of Bothnia.

- The authors state in the text that the modelled ice surface temperature shown in Figure 7 is for the configuration which doesn't include the snow scheme (i.e., SICE2D-NS), but it would be helpful to the reader if they also re-stated this in the figure caption.

**4  References**

- Bellouin, N., J. Rae, A. Jones, C. Johnson, J. Haywood, and O. Boucher, 2011: Aerosol forcing in the Climate Model Intercomparison Project (CMIP5) simulations by HadGEM2–ES and the role of ammonium nitrate, *J. Geophys. Res.*, **116**, D20206, doi:10.1029/2011JD016074.

- Hewitt, H. T., D. Copsey, I. D. Culverwell, C. M. Harris, R. S. R. Hill, A. B. Keen, A. J. McLaren, and E. C. Hunke, 2011: Design and implementation of the infrastructure of HadGEM3: The next-generation Met Office climate modelling system. *Geosci. Model Dev.*, **4**, 223–253, doi:10.5194/gmd-4-223-2011.

- Lea, D. J., I. Mirouze, M. J. Martin, R. R. King, A. Hines, D. Walters, and M. Thurlow, 2015: Assessing a New Coupled Data Assimilation System Based on the Met Office Coupled Atmosphere–Land–Ocean–Sea Ice Model. *Mon. Wea. Rev.*, **143**, 4678–4694, doi:10.1175/MWR-D-15-0174.1.

---

## Referee Comment (RC2) · Anonymous Referee #2 · 23 Apr 2018

General Comments:

This paper describes the impact of including a simplified ice scheme into the ALADIN-HIRLAM numerical weather prediction system version 38h1 for a short 2-month analysis period in March/April 2013 for 2 regions near Svarlbard and the Gulf of Bothnia with horizontal resolutions of 2.5 km. In the AROME Arctic domain, the SICE2D-NS (no snow) model performs best with the lowest mean error as a function of lead forecast time for mean sea level pressure, easily outperforming the reference run which does not include the SICE model. Some improvement is seen for that region with the 2-m air

temperature mean error, where up to a 0.5 °C improvement is made over the reference run where both have a negative bias. The SICE experiment without the snow model shows a fairly consistent 1° positive bias for the 3-45 hour lead forecast times. However, when examining the wind speed mean error, the reference run without the SICE model consistently showed the lowest bias. When examining the results for the Gulf of Bothnia domain, there was no discernible difference for any of the forecasts when examining sea level pressure. Overall, the SICE-NS experiment performed best for mean error and standard deviation for 2-m air temperature. The SICE-2D-S performed best for wind speed for both mean error and standard deviation. Experiments performed for March 2013 with a form drag parameterization (SICE-AD) could not outperform the reference run which did not include SICE. Qualitative figures presented for 2 days in March 2017 for the model versus MODIS ice surface temperature, bring little additional insight to the model performance.

The paper is filled with acronyms for numerous modeling systems (e.g., HARMONIE-AROME, ALADIN-HIRLAM etc.) which are never defined. The analysis period is short (March 1 – April 30, 2013), with limited data available for model-data comparisons.

For future reference, while coupling to an ice model such as CICE may provide the best overall improvement for Arctic NWP, the authors are encouraged to investigate the CICE Consortium's column physics package Icepack v1.0, which was released in February 2018. It is worth considering for future applications and is freely available to the public. (See https://github.com/CICE-Consortium/Icepack).

This paper is well written, but the study period is short. I recommend publication when the following issues are addressed in a revised version.

Specific Comments:

Although properly referenced, spell out all acronyms for the following:

HARMONIE-AROME, ALADIN-HIRLAM, CICE, GELATO, HIGHTSI, DWD, SURFEX

Section 3.3: The experiments occur during a short period of time, ∼2 months. Can these be extended for a longer period to better assess the model's performance? How does the model perform during the summer melt season? Instead of initializing with a constant ice thickness value of 0.75 m, consider testing with 28-day averaged near real-time, or seasonal values from CryoSat-2 (CPOM, see http://www.cpom.ucl.ac.uk/csopr/seaice.html). Seasonal (Spring and Autumn) derived ice thickness data is available going back to 2011). Data is available on a 1 and 5 km grid.

Page 13: Sec 3.4: I would like to see actual comparisons between MODIS and the model experiments (e.g., tabular statistics). On page 13 lines 7-8, you state "Statistical assessments require application of special methods, which were out of the scope of this study". Why? This would add value to your paper and possibly complement the results already shown. For the examples shown for March 2017, please add figures that show the HARMONIE-AROME run without SICE. In addition, check the availability of VIIRS ice surface temperature from NSIDC: https://nsidc.org/the-drift/data-update/viirs-sea-ice-surface-temperature-swath-data-now-available. If VIIRS is available, can you examine the difference between the modeled ice surface temperature with the SICE experiments versus the VIIRS product?

Technical Corrections:

Page 3 line 10: change to "The scheme that is developed"

Page 6 line 16: define ISBA

Page 6 line 25; insert a comma after "In this case"

Page 7 line 25: what do you mean by "screen level"?

Page 11, 12, 13 (twice), 15: replace "happens" with "occurs"

Page 15 line 25: should read "which is not available to the general public"

Page 20: Müller references should be listed as 2017a and 2017b; correct the text as necessary.

Page 20: check spelling for Posey references, several surnames spelled wrong.

Page 22: Is there a range for the number of snow layers? If yes, please state it.

Page 23: Table 2: Define the "ice scheme" in the caption.
* * *

---

## Author Comment (AC1) · 10 Jul 2018

Dear editor and referees,

thank you very much for the review of our manuscript. Please find the enclosed detailed response to your comments and suggestions as well as the marked-up version of the manuscript. Please note that our responses are marked by blue colour and indentation.

**Anonymous Referee #1**

1. General comments

   The paper presents results for the implementation of a simple sea ice thermodynamic model in a numerical weather prediction forecasting system. This is the first time such a sea ice model has been implemented in this particular NWP system, and a paper on the implementation and results should thus in principle merit publication.

   However, I have serious concerns about the use of a constant ice thickness, about the limited nature of the results used for validation of the model (covering only a short period in March-April 2013), and about some of the methods used for analysis. Some of the language used is also quite cumbersome and difficult to follow. I can therefore recommend publication only after major revisions, which I outline below.

2. Specific comments

   As mentioned above, I thought some of the language used was quite cumbersome and difficult to follow. I would strongly advise the authors to ask a native English speaker to proof-read the paper before resubmitting.

   > We have reworked some parts of our manuscript to make it more readable.

   The authors refer in the introduction to the Met Office Unified Model (page 2, lines 19-20), and state that "...to our knowledge there are no publications about the details of coupling between the advanced sea ice model and the atmospheric model in this system". This is incorrect. The model setup for coupled NWP is described by Lea et al. (2015), while the coupling is described in detail by Hewitt et al. (2011). The authors should cite both of these papers.

   > The paper by Lea et al. (2015) was cited in our manuscript couple of sentences earlier (page 2, line 18 of the first version of the manuscript). However in this paper there is no description of operational system, only of some experimental configuration. That it why we cite it to support the sentence about using coupled systems for research purposes. For the operational configuration of UK Met Office NWP system, from this paper it is clear only that SST is taken from OSTIA and kept constant during the forecast, but not clear what is happening with the ice surface temperature. The paper by Hewitt et al. (2011) provides many interesting details of coupling, but it describes the climate simulation system. In the new version of the manuscript, we cite the reference papers by Walters et al. (2017) and Rae et al. (2015) and correct the text to be more accurate in details: "In operational NWP they are applied in the global NWP systems to provide medium-range weather forecasts, e.g., in UK Met Office Unified Model (Walters et al., 2017; Rae et al., 2015) or IFS-ENS (Integrated Forecasting System–ENSemble prediction system, ECMWF, 2017). However, there are a number of reasons that advanced sea ice models are not widely used for short range operational NWP."

   On page 2, lines 15-18, the authors state that advanced sea ice models are "applied ... in coupled ocean-ice-atmosphere systems for research purposes and seasonal forecasting". I would suggest that they also mention that such sea ice models are used in coupled climate models, such as HadGEM3 (Hewitt et al., 2011).

*Corrected, as described in the response to the previous comment.*

The description of the model appears to be split between Sections 1, 2 and 3. How- ever, I would prefer to see one single model description section. This could be done in an expanded Section 2. In the current version of the paper, there is some description of the model setup in lines 5-27 of page 3 in the introduction. The authors then describe the sea ice parametrization scheme itself in Section 2. Then, within Section 3, Section 3.2 describes the experimental configuration. I appreciate that some level of discussion of the model is needed in the introduction, but I think the authors probably go into too much detail here. The introduction should set the scene, describe briefly the scientific background and work done by previous authors, and how the present paper builds on that. Much of the discussion of the model would more properly belong in a model description section. Similarly, Section 3 is primarily a results section. I think the description of the experimental configuration in Section 3.2 would again be better placed in an expanded Section 2.

*Corrected. Sentences related to model description are moved from Sections 1 and 3 to Section 2.*

The model is run with a constant ice thickness of 0.75m. However, the use of a constant thickness is likely to lead to be a source of considerable uncertainty, as ice surface temperature will be extremely sensitive to thickness. Indeed, the authors state in Section 3.1 that "when the ice thicknesses [in SICE and HIGHTSI] are very different, the ice surface temperature values may differ by more than 5°C", and in the conclusions, they say "the sensitivity of the results to the prescribed value of the ice thickness was noticed". Later, they also say that "the simplest way to go forward is to replace the prescribed constant value of the ice thickness by the climatology, to reproduce its seasonal and horizontal large scale variations". This begs the question of why the authors didn't use such a climatology in the current paper. In a resubmitted version of the paper, I would like to see results from a configuration of the model in which the local thickness in each gridbox is prescribed from climatology.

*We have run an additional experiment where the ice thickness from the model climatology derived from TOPAZ4 ice reanalysis is used and extended our manuscript accordingly.*

I am confused by Section 3.1. The authors state that they only compared SICE with HIGHTSI where the ice thickness in HIGHTSI was "approximately equal" to the constant thickness used in SICE (0.75m). However, they then state that they consider "small" ice thickness differences to be less than 0.4m, which is more than 50% of 0.75m; I would not say that such thicknesses are "approximately equal" to each other. The authors then state "When the ice thicknesses are very different, the ice surface temperature value may differ by more than 5°C", which suggests that, contrary to their previous statement, they have in fact analysed the results for larger differences in ice thickness between the two models. They also do not state what they mean by "very different" - do they mean the difference is greater than 0.4m?

*We guess that your confusion was arisen from quite an unclear wording in that paragraph. We did not intend to state that difference of 0.4 m in the ice ice thickness of HIGHTSI and SICE is small or large. We wanted to say that when ice thickness differ by more than 0.4 metres the ice surface temperatures computed by two models become considerably different.*

*We have reworded the problematic paragraph to make it more clear and avoid confusing assumptions about "small" or "large" differences in the ice thickness.*

In Section 3.2 (page 10, line 16), the authors mention that the model is "started from the snow-free state and allowed to accumulate snow from precipitation during the modelling period". What impact will this have on the forecasts? How long will SICE2D-S take to "spin up" to a realistic representation of the snow cover? It will surely be much longer than the 48 hours of the forecasts in the current paper. For this reason, I am not sure how much weight we can give to the SICE2D-S results. The authors should at least comment on this in the paper, and should preferably present results for SICE2D-S forecasts that have been started from a spun-up snow state.

Indeed, the SICE2D-S experiment started from a snow-free state, but it does not mean that each forecast started from a snow-free sea ice surface. HARMONIE-AROME experiments in this study have been run by using so-called cycling. Each cycle consists of data assimilation procedure and model forecast, and cycling period was 3 hours in this study. So, when HARMONIE-AROME starts a new cycle, it uses 3 hourly forecast results from the previous cycle as a background fields for data assimilation. But for sea ice variables there is no data assimilation, so sea-ice variables (e.g., the ice temperatures, snow water equivalent, snow-density, etc.) from the previous forecast are used, without any modifications, as an initial state for the current forecast. Therefore, in SICE2D-S experiment evolution of snow cover does not take just 48 hours, but goes throughout the whole experiment, that is 2 months.

To avoid further confusion we have added explanation to section 3.2 to show how snow on sea ice is initialized from cycle to cycle.

In Sections 3.3 and 3.4, the authors analyse forecasts only for a short period (March- April 2013). However, the performance of the model is likely to vary during the year, and the results may well be different in different seasons. I note that the authors state in the conclusions that in the future the model will be evaluated for more regions and more seasons, but I am of the opinion that results for other seasons must be included in the present paper for it to be worthy of publication. Given that Arctic sea ice exhibits a clear annual cycle, it is important that the impact of this on the performance of the model is analysed.

Yes we completely agree that selected study period is somewhat short and does not provide information about the performance of the ice scheme throughout the year. But when we test a high-resolution atmospheric model combined with a data assimilation system it becomes extremely costly in terms of CPU time, run time, storage, as well as in terms of cost of computing time on HPC to run experiments that are longer than just a couple of months.

We have no resources and funding that would allow us to run all experiments that were discussed in our study as a one year long experiment. However it is possible to use operational archive data to show the performance of the ice scheme during different seasons, selecting the ice surface temperature as a verification variable. In this case additional experiments are not needed.

To show the performance of the new sea ice scheme throughout the year we have compared ice surface temperatures from the operational AROME Arctic and near real time L2 ice surface temperature values derived from MODIS and VIIRS sensors that cover the time period from December 1st 2015 to December 1st 2016. To study the performance of HARMONIE-AROME without SICE scheme throughout the year, the ice surface temperature fields interpolated from the host model IFS-HRES were used (because they would be used in the forecast without SICE scheme). We have extended Section 3.4 accordingly to show the results of this new comparisons.

In Figures 2, 3 and 6, the authors present results for the mean error in the forecast MSLP, 2-metre temperature, and 10-metre wind speed, where the error is defined as the difference between the modelled and observed quantities, and the mean is taken over several observing sites. However, the standard deviations plotted in Figures 2, 3 and 6 are often much larger than the differences between the means. The authors discuss the differences between the mean errors for different experiments at length in Section 3.3, and consider possible reasons for them, but the fact that the standard deviations are so large compared to the differences suggests that the differences may not be significant. If this is indeed the case, then it suggests that the SICE scheme, and the related snow and form drag schemes, may not have a significant impact on the model-obs errors. It would be interesting to see if this conclusion changes when the authors use RMS error rather than simple mean error, and when they look at forecasts for different times of year.

This question indeed contains three aspects: 1) statistical significance,

2) relation between bias, error standard deviation and root-mean-square error, and

3) forecast errors for different times of the year. Aspects (2) and (3) are also mentioned in your other comments, so we discuss them in the answers to these comments. Aspect (1) we discuss here. We compare forecast errors between different numerical experiments for the forecasts of different length. We consider the forecast error as a random value. So, our sample for calculating statistics has the following volume: $(number\ of\ forecasts\ per\ day) * (number\ of days) * (number\ of\ stations)$. For example, for Figure 2 in the previous version of the manuscript, the sample volume is $(1 forecast\ per\ day) * (60 days) * (7 stations) = 420$ cases. This is quite large sample volume, even if we consider only 7 stations – because we have 30 days to make statistics. With samples of these volume, even with large standard deviations, the results are usually statistically significant. In this example, for the 39-hour forecasts, the difference in the mean errors (biases) between experiments REF and SICE2D-NS is statistically significant for 2-metre temperature and MSLP with the level of 0.1%, and for 10-metre wind with the level of 5%. Usually in NWP the sample volumes are large or very large, sometimes thousands of cases. That is why the mathematically strict statistical significance estimates are often not displayed on verification plots in NWP. In our case, the sample volume may be sometimes lower then hundreds of cases. Thus, in the new version of the manuscript, although we don't estimate the statistical significance strictly, but we discuss the sample volume and make circumspect conclusions in cases when it is small.

Another point relating to Figures 2, 3 and 6 is that the use of a simple mean error will potentially lead to positive and negative errors cancelling each other out. This will hide potentially-relevant results if some stations have a very large positive bias and others have a large negative bias. For this reason, I think the root-mean-square error would be a more useful quantity to assess, and I would like to see a plot of this, either instead of or in addition to the simple mean error that the authors have plotted here.

In fact, all three errors, namely the mean error (BIAS), the error standard deviation (ESTD) and the root-mean-square error (RMSE) are connected with each other. Indeed, if FE is the forecast error, then BIAS is an average FE, ESTD is a SQRT of an average of (FE-BIAS)^2, RMSE is a SQRT of an average of FE^2, then

RMSE^2=BIAS^2+ESTD^2

That is why, we may use either RMSE or ESTD to estimate the random error of the forecast. With BIAS, we estimate a systematic error, so BIAS is also important. RMSE, in turn, contains both: BIAS and ESTD. Positive and negative errors of different forecasts, although cancelling each other in BIAS, are reflected in ESTD, not only in RMSE. But RMSE is very traditional, so we add it. Also, estimates of statistical significance of the difference of RMSE between two experiments is problematic, because in this case the random value is FE^2, which has non-gaussian distribution. Then, Student's criterion is not applicable, and more complicated methods are necessary.

It would also be interesting to see the contributions of the different observing stations to the mean (or RMS) error. This could be done using maps of a similar form to Figure 3 of Bellouin et al. (2011), where the observations are shown with boxes superimposed on a map showing the fields output by the model.

Thank you for such a good idea! We have added a map showing the relative change in 2 metre air temperature RMSE of the SICE2D-NS experiment compared to the REF experiment (See Figure 2 in the updated manuscript).

This figure also shows that stations with the most noticeable difference in RMSE between the SICE2D-NS and REF experiment are located in coastal areas close to the sea ice and for the inland stations there is almost no differences in RMSE between the SICE2D-NS and REF experiments.

The authors mention in the text (page 10, line 21) that they used 12 Svalbard stations, and indeed 12 are shown on the map in Figure 1. However, in the captions of Figures 2 and 6, they mention "7 Svalbard stations", and list the 7 stations. I presume this is because of the issues described in Section 3.3 (page 10, lines 23-26) whereby some Svalbard stations were excluded because they were in fjords. But were the other 5 stations used at all in this analysis? If not, then it is incorrect to state at the beginning of Section 3.3 that measurements from 12 Svalbard stations were used (as in fact only 7 were used). The authors should re-word this paragraph (page 10, lines 21-26) to make this clearer.

> Yes, you are right. From the 12 selected stations only 7 were surrounded by sea ice and the rest of them was close to the open sea or located in fjord areas where interpolation issues were noticed.

> To avoid confusion, we have reworded description of selected Svalbard stations to clearly show that some of stations were removed. We have also updated the Figure 1 to show only the stations that were actually used for the comparisons.

In Section 3.3 (page 11, lines 14-18), the authors discuss the relative sizes of the standard deviations of the errors in REF and SICE2D-NS, without any mention of the implications or relevance of these results. Presumably a smaller standard deviation implies that there is a smaller range of errors between stations. Is this relevant, and if so why? Is there any indication what might be causing it? Does the implementation of the sea ice scheme affect the 2-metre temperature at some stations more than others? Is there an obvious reason for this?

> Indeed, the smaller error standard deviation indicates that the range of 2 metre temperature (T2m) errors is smaller in SICE experiments than in REF. In the other words, random component of the modelling errors has been reduced. This is a desirable property for an operational NWP system because systematic errors of a forecast could be corrected later during the post-processing stage.

> When using SICE, sea ice cover is defined by the ice concentration field and ice surface temperature is computed by the ice scheme. But in REF ice cover is taken from the surface temperature forecast by IFS-HRES and kept constant during the forecast. As result, in REF, constant ice surface temperature filters out variations in T2m and leads to considerable range of errors for different SYNOP stations.

> Of course, the effect of SICE on the T2m forecast varies from stations to station. This is because the T2m forecast could be strongly dependent on the local conditions, such as characteristic wind direction or station elevation, notwithstanding the fact that stations surrounded by closed ice would show more clear response than stations that have only traces of ice in their vicinity.

> We have extended the section 3.3 to make the text more clear and indicate that errors in SICE experiments are less random than in REF.

At the end of Section 3.3 (page 12, lines 31-34), the authors state "...with observations from coastal stations only, we lack understanding of the ice temperature behaviour for larger scales". This is a very good point to make, and I would recommend that when the authors resubmit the paper they include results for a wider range of stations within the forecast domain, including non-coastal (i.e. inland) stations. Does the implementation of the sea ice scheme affect the results only at stations that are physically close to the sea ice, or are there larger-scale effects?

> When stating "...with observations from coastal stations only, we lack understanding of the ice temperature behaviour for larger scales" we meant the characteristics of the ice field itself rather than performance of the inland stations. Inland stations that are located hundreds and thousands kilometres away from the sea ice show almost no response to changes in the sea ice parameterization, as can be seen from the provided map in the updated version of manuscript. And to study the larger-scale performance over the sea ice covered areas we have compared results of numerical experiments and extended our manuscript accordingly.

> We have provided scatter plots of relative change in RMSE induced by SICE to show the impact over all available weather stations.

Figure 7 shows surface temperature derived from MODIS data, and forecast by the model. However, it is quite difficult to get an idea of the differences between the temperature fields in the plots. It would be much more useful if the authors could present maps showing the difference between these (i.e., model minus MODIS). This would help the reader to understand better the results discussed in Section 3.4.

> Figure 7 was meant to illustrate the similar patterns in the ice surface temperature fields from SICE within the operational NWP system and MODIS ice surface temperature product. We have replaced it by the maps of root means square errors of the ice surface temperature. We provide such figures for short-term experiments, discussed in the Section 3.3 and for evaluation of performance of the operational NWP system which uses SICE for parameterization of sea ice, discussed in the Section 3.4.

The authors mention in Section 3.4 (page 13, lines 10-12) that most MODIS swaths were in the daytime. However, the model results shown in Figure 7 are whole-day averages. How will this affect the comparison of the two? I imagine there may be a warm bias in the MODIS observations as a result of the fact that they are generally restricted to daytime. The authors should comment on this, and its implications for the results, in the paper.

> Thank you for this comment. We agree that daily aggregated ice surface temperature product should not be compared to time-averaged model results. To make a more clear comparison in the new version of the manuscript we use near real rime ice surface temperature products instead of daily aggregated ones. To show the difference between the forecasted ice surface temperature and data provided by satellite products we have replaced Figure 7 by maps of the root mean square error for different experiments (see our reply to the previous comment).

3. Technical corrections

   – Page 2, lines 2-3: "Over areas with a mixture of floes and polynyas, the form drag appears, which affects the turbulent fluxes". I would suggest re-wording this, so that it reads: "Over areas with a mixture of floes and polynyas, the turbulent fluxes are affected by form drag".

   > Reworded according to the suggested variant

   – Page 2, lines 19 and 34: I don't like the use of "To our knowledge. . . ", as it seems unscientific to me. I have already mentioned above that the statement made on lines 19-20 is in fact incorrect. I would also suggest an alternative wording for the sentence on lines 34-35. Indeed, if one doesn't know whether a particular statement is true or not, it is often best not to include it at all, rather than preceding it with "To our knowledge. . . ".

   > We have reworded those sentences to avoid unscientific language.

   – Page 4, line 6: ". . . it is designed so that it can be naturally coupled with a snow scheme. . . ": I don't know what the authors mean by "naturally coupled". I think that ". . . so that it can be coupled to a snow scheme. . . " would suffice.

   > Reworded according to the suggested variant

   – Page 7, line 25: "It is important to mention that. . . ": This is unnecessary. If it's important to mention it, then mention it – there is no need to say that it's important to do so.

   > We have removed "It is important to mention that. . . " as suggested.

– Page 9, line 25: "The background for the data assimilation are fields of prognostic variables. . . ": I think there is a word missing here, and that this should read " The background fields for the data assimilation. . . ".

Fixed

– Page 10, line 27: Figure 6 is mentioned before Figures 4 and 5. The figures should be re-ordered to avoid this.

We have re-ordered figures to avoid this situation

– Page 11, lines 6-7: ". . . the underestimation of night-time 2 metre temperatures over land is a characteristic feature of the model known from operational verification (not shown)". If this is known from operational verification, is there a reference that the authors can cite?

That sentence was supposed to be removed from the submitted manuscript and was left there by a mistake. We have removed that sentence and added explanation why there is a noticeable diurnal cycle of the 2 metre temperature bias in the REF experiment.

– Page 11, lines 14-15: "The error standard deviation for the 2 metre temperature forecasts. . . ". This should read "The standard deviation of the errors in the 2 metre temperature forecasts. . . ".

Reworded according to suggested variant

– Page 11, line 22: "mean sea level pressure error standard deviation" sounds clumsy. I would suggest "standard deviation of the error in mean sea level pressure". The authors could also abbreviate "mean sea level pressure" to "MSLP", if they define the abbreviation the first time they use it.

That phrase has been reworded according to suggestions

– Page 11, line 31: " over the part of the grid cell related to the sea with ice". I'm not sure what this means. Does it mean "over the ice-covered part of the grid cell", or something else? I would suggest re-wording this to make it clearer.

This sentence supposed to mean "over the part of grid cell that contains both open sea and sea ice". We have reworded that sentence to make it less confusing.

– Page 12, lines 4 and 18: I think the authors mean "in agreement with" rather than "in accordance with".

Fixed

– Page 12, line 20: ". . . makes the surface temperature drop more and more". This language ("more and more") is not very scientific. Please consider re-wording.

We have reworded that sentence to use more scientific language

– Page 14, line 18: ". . . the sensitivity of the results to the prescribed value of the ice thickness was noticed". I think the authors mean "noted" rather than "noticed".

Fixed

– In the caption of Figure 3, the authors mention 7 stations in the Gulf of Bothnia, and 7 are shown in the map in Figure 1, but in the text (page 10, line 21) they state that they used 6 stations in the Gulf of Bothnia.

Information about the number of used stations in the Gulf of Bothnia has been corrected in the text.

– The authors state in the text that the modelled ice surface temperature shown in Figure 7 is for the configuration which doesn't include the snow scheme (i.e., SICE2D-NS), but it would be helpful to the reader if they also re-stated this in the figure caption.

> Figure 7 has been removed from the updated version of the manuscript, but we have extended the corresponding captions for the new figures to explicitly state that these results were obtained from the snow-free configuration of SICE.

From the statistical point of view, model errors obtained from the comparison with satellite observations are in fact time series of random 2D fields. Thus, the verification statistics will be dependent on the methods of sampling (in time, or in space, or both), of aggregation the information from one grid to another, etc. At least, this is a huge amount of data to process. Usually these kind of studies deserve special attention and special publications. But we agree that adding comparisons between satellite products and model experiments would add value to our study and we have extended Section 3.4 to provide this information. Please see our answer to the similar question, raised by the first Referee.

Technical Corrections:

Page 3 line 10: change to "The scheme that is developed"

This sentence has been removed from the updated version of manuscript

Page 6 line 16: define ISBA

Added

Page 6 line 25; insert a comma after "In this case"

Fixed

Page 7 line 25: what do you mean by "screen level"?

The term "screen level" corresponds to the mount height of sensors, situated inside the thermometer screen, which should be within the range of 1.25 to 2 metres according to Guide to Meteorological Instruments and Methods of Observation (WMO, 2008). To avoid further confusion we have rearranged text to avoid mixing screen level parameters and other meteorological parameters such as snow depth or 10 metre wind speed.

Page 11, 12, 13 (twice), 15: replace "happens" with "occurs"

Fixed

Page 15 line 25: should read "which is not available to the general public"

Changed according to recommendation

Page 20: Müller references should be listed as 2017a and 2017b; correct the text as necessary.

We have corrected the text to list Müller references as 2017a and 2017b.

Page 20: check spelling for Posey references, several surnames spelled wrong.

Spelling has been corrected

Page 22: Is there a range for the number of snow layers? If yes, please state it.

We have added the technically valid range of the number of snow layers. The number of snow layers for ISBA ES in SICE can not be changed through the configuration file, but should be done directly in the source code.

Page 23: Table 2: Define the "ice scheme" in the caption.

Caption of Table 2 has been extended to provide definition of the "ice scheme" column.

[revised manuscript text omitted]

---

## Author Response (AR2)

Dear topical editor,

please find the enclosed detailed response to your comments as well as the marked-up version of the manuscript. Please note that our responses are marked by blue colour and indentation.

Comments to the Author: On the whole this is a well thought out and thorough revision to the paper.

The concern that has not really been completely addressed is the shortness of the run used for evaluation. I am confused as to why it is impossible to run the model for longer, as I thought that this model is to be part of the operational system, which means there must be resources to run it for longer. Please explain this.

> A full 2-month experiment of HARMONIE-AROME in configuration that was used in our study takes approximately 1 month of real time for calculations. In the research mode model runs are much slower (because of limited number of available processing nodes) than in the operational one. Regarding that the alternative is no sea ice scheme at all, the positive results from these relatively short spring experiments were considered by the Norwegian Meteorological Institute to be enough to make a decision to run SICE within the operational system where its performance is monitored on the daily basis by forecasters.

> After running SICE operationally for a year, archives became available for validations. We used these archives to analyze the performance of SICE throughout the year in the manuscript.

> In the first revised version of our manuscript we provided the results of evaluation of one year of operational runs with SICE against the MODIS and VIIRS ice surface temperature products. We presume that these results didn't draw too much attention. In the new version of manuscript we have re-arranged text to make it more clear that we use a series of relatively short numerical experiments to evaluate the effect of using the SICE scheme and we assess the performance of SICE within the operational system for a one-year period.

Related to this, it is not clearly stated in the manuscript (at least I could not find it) why such a short period has been chosen. In the case where a longer run really is a computational impossibility due to lack of resource, this must be stated in the manuscript, as it is not reasonable to expect readers to go trawling through the response to the referees to find this out. Also, please explain in the manuscript why was this particular snapshot of history was chosen. It might also be helpful to admit that this is a limitation of the work and explain what evaluation experiments are expected to be performed in future.

> Please see our response to the previous comment for explanation about the length of numerical experiments.

> For the SICE2D experiments, the early spring season was chosen because during this part of the year the polar night is already over, but still it is cold enough, so that the sea ice temperature has well pronounced diurnal cycle. New experiments would add the value to validations, because in addition to remote-sensing observations they will allow using SYNOP data and perhaps other data. But now they are not so urgent. We hope that after some restructuring of the text this situation became more clear. We explain the reason of choosing the early spring season for the SICE2D experiments in the text. Also, we have added more plans for evaluations to the Conclusions.

In relation to the model-data differences, I agree that it is useful to split the RMSE into mean and STD, but do not agree with you calling the STD the "random" component of the error. The STD is the variation about the mean. There is nothing random about it.

> Actually, behind analyzing forecast errors is the following. We consider the forecast error (the delta between the forecast and the observation value) as a random variable. Our forecast value is actually not random, because we calculate it. But observation value may be considered as random. Then, this random variable (the forecast error) has some probability distribution function. Very often (but not always) this function is assumed to be Gaussian. Then we calculate the statistics of this function. When calculating statistics, we use the ergodic hypothesis because in the case of deterministic forecast we have no ensemble

of realizations of the random variable, but only observations in different times and locations. Statistics are the mean value (the first statistical moment) and the variance (the second statistical moment). From the variance, we calculate the standard deviation as square root. We call the mean value of the forecast error as BIAS, end the standard deviation as ESTD. Root-mean-square error (which is calculated using the forecast error squared) actually contains both BIAS and ESTD. Then, we agree, that mathematically, we can't say that "BIAS is a systematic component of our forecast error and ESTD is it's random component", because both of them are just *statistics* of a *random variable*. But physically this interpretation is valid due to the ergodic hypothesis and often used. Still, to be more mathematically strict, we have corrected the appropriate phrases in the new version of the manuscript.

It would be too cruel to say that the language is tortured, but it is still someway from the elegant simplicity that enables easy reading. Copy editing will improve it further, but may not be able to do much with long and twisted sentences. Please try and make some further improvements to this in the next version.

We have edited our manuscript to improve its readability. To make the structure of manuscript more clear we have re-organized the Section 3.

The geoscience community is very large, and it is not possible for us all to know everything. It is therefore perfectly fine to say things like, "to the best of our knowledge" rather than to make a statement of which you are not confident. In particular, "To our knowledge, no other observations of ice properties are used for assimilation in short range NWP systems" has become, "Other observations of ice properties are rarely used for assimilation in short range NWP systems". This change suggests that you now have knowledge that ice properties are sometimes used, in which case you should give references! Instead, I suggest that this is perfectly acceptable: "The authors of this paper know of no other cases where observations of ice properties are used for assimilation in short range NWP systems".

We have mentioned the microwave emissivity of sea ice derived from satellite measurements as an example of additional observations of sea ice properties that are used in NWP.

[revised manuscript text omitted]